# UniSplat: Unified Spatio-Temporal Fusion via 3D Latent Scaffolds for Dynamic Driving Scene Reconstruction

**Chen Shi**[1,3]    **Shaoshuai Shi**[3]    **Xiaoyang Lyu**[2,3]
**Chunyang Liu**[3]    **Kehua Sheng**[3]    **Bo Zhang**[3]    **Li Jiang**[1†]

[1]The Chinese University of Hong Kong, Shenzhen
[2]The University of Hong Kong    [3]Voyager Research, Didi Chuxing

**Project Page:** `https://chenshi3.github.io/unisplat.github.io/`

## ABSTRACT

Feed-forward 3D reconstruction for autonomous driving has advanced rapidly, yet existing methods struggle with the joint challenges of sparse, non-overlapping camera views and complex scene dynamics. We present UniSplat, a general feed-forward framework that learns robust dynamic scene reconstruction through unified latent spatio-temporal fusion. UniSplat constructs a 3D latent scaffold, a structured representation that captures geometric and semantic scene context by leveraging pretrained foundation models. To effectively integrate information across spatial views and temporal frames, we introduce an efficient fusion mechanism that operates directly within the 3D scaffold, enabling consistent spatio-temporal alignment. To ensure complete and detailed reconstructions, we design a dual-branch decoder that generates dynamic-aware Gaussians from the fused scaffold by combining point-anchored refinement with voxel-based generation, and maintain a persistent memory of static Gaussians to enable streaming scene completion beyond current camera coverage. Extensive experiments on real-world datasets demonstrate that UniSplat achieves state-of-the-art performance in novel view synthesis, while providing robust and high-quality renderings even for viewpoints outside the original camera coverage.

## 1 INTRODUCTION

Replicating 3D scenes from urban driving sequences has emerged as a core capability for autonomous systems, supporting simulation (Cao et al., 2025; Yang et al., 2023; Tonderski et al., 2024), scene understanding (Huang et al., 2024b; 2025; Yan et al., 2025a), and long-horizon planning (Murai et al., 2025). Recent advances in 3D Gaussian Splatting (Yan et al., 2024; Jiawei et al., 2025; Kerbl et al., 2023) have demonstrated impressive rendering efficiency and fidelity. However, these methods typically assume substantial viewpoint overlap among input images and rely on per-scene optimization, which limits their applicability in real-time driving scenarios.

To enable faster inference, feed-forward reconstruction methods have emerged to synthesize novel views in a single forward pass (Xu et al., 2025; Chen et al., 2024; Zhang et al., 2025; Lu et al., 2024). These methods typically encode inter-view correlations within the image domain via cross-attention or by constructing a multi-view stereo (MVS) cost volume, and subsequently decode the Gaussian primitives from the resulting fused features. Notably, the choice of fusion strategy is crucial, as it significantly impacts the final rendering quality. EvolSplat (Miao et al., 2025) integrates multi-frame geometric information from front-view monocular sequences using 3D-CNN, but ignores semantic fusion and lacks mechanisms for dynamic handling. Meanwhile, Omni-Scene (Wei et al., 2025) leverages a Triplane Transformer for strong multi-view fusion but does not incorporate temporal

---

†: Corresponding author.

aggregation and is constrained by coarse-grained 3D details. Despite these advances, robust reconstruction in urban driving scenarios remains challenging, particularly in maintaining a unified latent representation that evolves smoothly over time, handling partial observations, occlusions, and dynamic motion, and efficiently generating high-fidelity Gaussians from sparse inputs.

To address these challenges, we propose **UniSplat**, a general feed-forward framework for dynamic scene modeling from multi-camera videos. The core insight of UniSplat is to construct a unified 3D scaffold that fuses both multi-view spatial information and multi-frame temporal information. This scaffold facilitates geometric and semantic contextual interaction in 3D space, supports efficient long-term information integration and dynamic modeling, and enables effective decoding of Gaussian primitives. By preserving and fusing essential information, it ensures coherent and consistent scene reconstruction over time.

Specifically, the UniSplat framework follows a three-stage pipeline. First, we construct an egocentric 3D scaffold by feeding multi-view images to a pretrained geometry foundation model and a visual foundation model, encoding both geometry structure and semantics cues into a sprase 3D feature volume. Second, we perform spatio-temporal fusion by integrating multi-view spatial context within the current frame's scaffolds and fusing historical scaffolds into current scaffolds via egomotion compensation, yielding a temporal-enhanced scene representation. Third, we decode the fused scaffold into Gaussians via a dual-branch strategy: one branch predicts Gaussians at sparse point locations for fine-grained detail while the other directly generates new Gaussians from voxel centers to complement point anchor predictions. Each Gaussian is assigned a dynamic probability score to identify static content, allowing us to maintain a memory bank of persistent static Gaussians across frames for long-term scene completion.

We evaluate our method on the Waymo Open dataset (Sun et al., 2020) and NuScenes (Caesar et al., 2020) dataset, which present dynamic street scenes with complex environmental conditions and limited overlap for multi-camera images. Experimental results demonstrate that our approach achieves state-of-the-art performance across both datasets in input-view reconstruction and novel-view synthesis. Notably, with the help of temporal memory, our model exhibits strong robustness and superior rendering quality when synthesizing views outside the original camera coverage.

In summary, our main contributions are as follows:

- We introduce UniSplat, a novel feed-forward framework for dynamic scene reconstruction from multi-camera videos via a unified 3D latent scaffold.

- We design a novel scaffold-based fusion mechanism that supports unified spatio-temporal alignment and progressive scene memory integration.

- We propose a dual-branch Gaussian generation mechanism with dynamic-aware filtering, enabling fine-grained and complete rendering and memory-based scene completion.

- Comprehensive experiments on two large-scale driving datasets demonstrate that UniSplat significantly outperforms state-of-the-art feed-forward reconstruction methods, with generalization capability for challenging views outside the observed camera frustums.

## 2 RELATED WORK

**Neural 3D Reconstruction.** The field of neural 3D reconstruction has witnessed remarkable progress, largely driven by Neural Radiance Fields (NeRF) (Mildenhall et al., 2021) and, more recently, 3D Gaussian Splatting (3DGS) (Kerbl et al., 2023). NeRF represents scenes as continuous volumetric functions, achieving high-fidelity renderings but incurring substantial computational costs. Subsequently, 3DGS introduced explicit point-based representations with highly efficient rasterization, enabling real-time rendering. Despite the impressive performance of NeRF, 3DGS, and their extensive variants (Hu et al., 2023; Xu et al., 2022; Müller et al., 2022; Hu et al., 2023; Yu et al., 2024; Yang et al., 2025a), these methods are usually limited by the reliance on dense input views and costly per-scene optimization, thereby restricting their scalability. Alternatively, feed-forward methods tackle this challenge by learning generalizable scene priors from large-scale datasets during training, allowing for immediate reconstruction from sparse observations at inference time. MuRF (Xu et al., 2024) employs target view frustum volumes for radiance field reconstruction. PixelSplat (Charatan et al., 2024) and Splatter Image (Szymanowicz et al., 2024) predict per-pixel

3D Gaussians from image features, while MVSplat (Chen et al., 2024) leverages cost volumes for geometric consistency and DepthSplat (Xu et al., 2025) integrates features from pre-trained monocular depth models to improve robustness. However, these approaches still face significant challenges in complex urban driving scenarios, where minimal overlap among surround-view cameras compromises multi-view correspondence and the presence of highly dynamic objects complicates temporal aggregation. Beyond these explicit geometric methods, token-based transformers (Jin et al., 2025) and diffusion-based models (Gao et al., 2025; 2024) have also been explored for generalizable view synthesis without explicit reconstruction, but they typically suffer from high rendering costs or hallucinate content that is inconsistent with the input context. In this work, we develop a feed-forward framework to reconstruct complete driving scenes from sparse views while effectively leveraging multi-frame information.

**Driving Scene Reconstruction with 3D Gaussians.** Leveraging advances in 3D Gaussian Splatting, several works (Chen et al., 2023; Huang et al., 2024a; Zhou et al., 2024b; Yan et al., 2024; Zhao et al., 2025; Yan et al., 2025b; Fan et al., 2025; Jiawei et al., 2025) specialize in driving scenes (Pei et al., 2025a;b), focusing on 3D or 4D reconstruction within individual scenes through offline optimization. In parallel, generalizable methods have emerged. These approaches (Tian et al., 2025; Lu et al., 2024) typically employ depth networks to determine Gaussian primitive positions in a feed-forward manner and predict per-pixel Gaussians along camera rays. To enhance global consistency and completeness, several techniques incorporate 3D spatial representations. EVolSplat (Miao et al., 2025) directly accumulates depth across multiple frames and leverages 3D-CNNs to refine Gaussian geometry. Omni-Scene (Wei et al., 2025) transforms multi-view image features into Tri-Plane representations and decodes voxel-anchored Gaussians to complement pixel-based estimates. SCube (Ren et al., 2024) constructs a detailed sparse-voxel scaffold via a hierarchical voxel latent diffusion model. However, these methods often focus on static or single-frame reconstruction and struggle to simultaneously handle multi-view fusion and dynamic multi-frame aggregation. More recently, unsupervised 4D reconstruction approaches have been proposed, but they either lack effective 3D alignment for complex scene flow estimation (Yang et al., 2025c) or require LiDAR initialization (Wang et al., 2025b). To counter these challenges, we propose UniSplat, a novel framework that unifies multi-view fusion and dynamic multi-frame aggregation within a 3D latent scaffold.

**3D Geometry Reconstruction.** End-to-end, data-driven pipelines that reconstruct scene geometry directly from images have progressed rapidly. DUSt3R (Wang et al., 2024) pioneers a transformer-based framework that predicts 3D point maps from uncalibrated image pairs. Subsequent works (Wang et al., 2025f;a; Yang et al., 2025b; Wang et al., 2025c; Chen et al., 2025; Xiao et al., 2025) extend this paradigm to arbitrary multi-view inputs and scale up both training data and model capacity, achieving state-of-the-art reconstruction accuracy with strong generalization across diverse scenes. However, these methods generally struggle with poor texture representation and encounter multi-view misalignment under minimal overlap, limiting novel view rendering quality. In this work, we employ these 3D foundation models to obtain a geometry initialization from images, and then perform 3D alignment and fusion in the learned latent scaffold.

## 3 UNISPLAT

UniSplat operates on a continuous stream of multi-camera frames, maintaining a unified 3D latent representation of the scene that evolves over time. As shown in Fig. 1, each time step begins with 3D scaffold construction from multi-view images (Sec. 3.2), producing a set of 3D voxels (the latent scaffold) that encodes the scene's geometry and semantics in an ego-centric coordinate frame. We then perform a unified spatio-temporal fusion, integrating information across views within the current scaffold and aggregating it with the latent scaffold from the previous time step (Sec. 3.3). Finally, we achieve dynamic-aware Gaussian generation (Sec. 3.4) through a dual-branch decoder that estimates dynamic-aware Gaussian primitives from both points and voxels, while maintaining a temporal memory bank that accumulates static Gaussians over time to address incomplete scene coverage caused by sparse camera inputs and limited fields of view.

### 3.1 PRELIMINARY

3D Gaussian Splatting (Kerbl et al., 2023) represents a scene as a collection of 3D Gaussian primitives $\mathcal{G} = \{G_i\}_{i=1}^{N}$. Each primitive $G_i$ is defined by a tuple of learnable parameters

$\theta_i = \{\boldsymbol{\mu}_i, \alpha_i, \boldsymbol{\Sigma}_i, \mathbf{c}_i\}$, representing its 3D center position, opacity, covariance matrix, and color coefficients, respectively. To render an image from a target viewpoint, these 3D Gaussians are projected onto the 2D image plane and blended using differentiable alpha compositing. Specifically, for a particular pixel, the color contribution $C$ from all Gaussians whose projections cover that pixel is:

$$C = \sum_{i \in \mathcal{N}} \mathbf{c}_i \alpha_i \prod_{j=1}^{i-1} (1 - \alpha_j), \tag{1}$$

where $\mathcal{N}$ is the set of Gaussians overlapping the pixel, sorted by depth. Beyond simple color rendering, several works (Zhou et al., 2024a; Zuo et al., 2025) augment Gaussians with additional parameters, which can be rendered into a 2D feature map using the same alpha compositing mechanism, enabling the distillation of knowledge from 2D foundation models. Inspired by this extensibility, we introduce a learnable dynamic attribute for each Gaussian to explicitly disentangle scene dynamics.

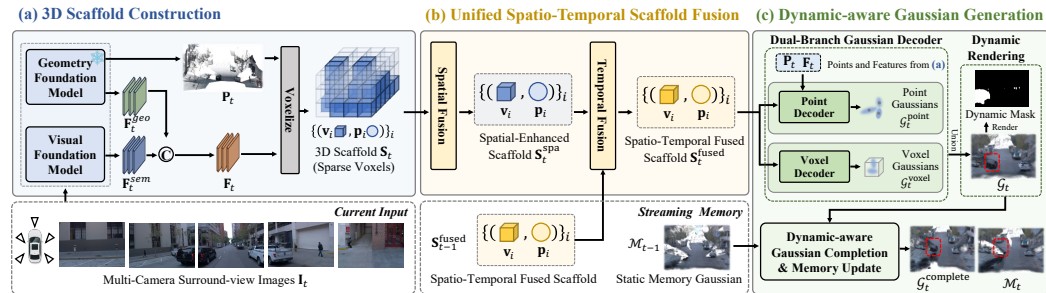

Figure 1: **Overview of UniSplat.** Given multi-camera images from vehicle-mounted cameras, UniSplat leverages foundation models to construct geometry-semantic aware 3D latent scaffolds, where unified spatio-temporal fusion is performed. From this scaffold, a dual-branch decoder generates dynamic-aware Gaussian primitives using both point anchors and voxel centers, with dynamic filtering maintaining a persistent memory of static scene content. The red boxes highlight a dynamic car that is filtered out in our memory module (best viewed when zoomed in).

## 3.2 3D SCAFFOLD CONSTRUCTION

Constructing an accurate 3D scaffold from sparse, minimally overlapping camera views is a primary challenge in multi-view reconstruction for driving scenes. To address this, we harness the power of geometry foundation models to infer a coherent 3D structure from multi-view images in one forward pass. We then enrich this 3D geometric scaffold with semantic information from a visual foundation model. This process yields a latent scaffold representation in the ego-centric coordinate frame of the vehicle, which provides a strong basis for subsequent spatio-temporal fusion.

**Metric-Scale 3D Geometry Generation.** Given synchronized multi-view images $\mathbf{I}_t = \{I_t^k\}_{k=1}^{N_{\text{cam}}}$ from a multi-camera rig, we apply a state-of-the-art feed-forward multi-view geometry foundation models (e.g., Wang et al. (2025a;f)) to directly predict a dense 3D point map $\mathbf{P}_t^{\text{init}} \in \mathbb{R}^{N_{\text{cam}} \times H_{\text{img}} \times W_{\text{img}} \times 3}$, where each pixel is associated with a 3D coordinate inferred jointly from all views. Unlike per-view depth estimation and late fusion, this unified approach leverages learned multi-view correspondences to generate a coherent scene-level point cloud. However, such predictions often suffer from scale ambiguity, which is problematic in autonomous driving. Thus, we introduce an auxiliary scale alignment branch: a small MLP predicts per-camera scale factors from the pooled geometry features:

$$\boldsymbol{\gamma} = \text{MLP}(\text{AvgPool}(\mathbf{F}_t^{\text{geo}}, \{H, W\})) \in \mathbb{R}^{N_{\text{cam}}}, \tag{2}$$

where $\mathbf{F}_t^{\text{geo}}$ denotes the hidden feature maps from the geometry model and $\text{AvgPool}(\cdot, \{H, W\})$ represents the average pooling over the height and width dimensions for each view. The scale prediction is supervised by minimizing the error between $\boldsymbol{\gamma}$ and the optimal scale vector computed using the ROE solver (Wang et al., 2025d) with LiDAR point references. Applying $\boldsymbol{\gamma}$ to $\mathbf{P}_t^{\text{init}}$ yields a metric-consistent point cloud $\mathbf{P}_t$ as the geometric foundation of our scaffold.

**Scaffold Construction with Geometric-Semantic Context.** As the generated $\mathbf{P}_t$ is an unstructured point set, we organize these points into a sparse voxel grid and fuse geometric and semantic

features to create the 3D latent scaffold. To achieve this, we first extract semantic-aware 2D features $\mathbf{F}_t^{\text{sem}}$ from the input views using a visual foundation model (Oquab et al., 2023), and fuse them with geometric features $\mathbf{F}_t^{\text{geo}}$ to obtain a unified multi-view feature map $\mathbf{F}_t$. We then voxelize the point cloud into $N_v$ voxels within an ego-centric cuboid $[\mathbf{p}_{\min} \in \mathbb{R}^3, \mathbf{p}_{\max} \in \mathbb{R}^3]$ covering the surrounding scene. The volume is partitioned into voxels of size $\epsilon$, and only voxels containing points are considered valid. Specifically, for each voxel $i$, we compute its coarse geometric voxel feature $\mathbf{v}_i^{\text{init}}$ as the average of the coordinates of points $j \in \mathcal{I}_i$ that lie in that voxel:

$$\mathbf{v}_i^{\text{init}} = \frac{\sum_{j \in \mathcal{I}_i} \mathbf{P}_{t,j}}{\sum_{j \in \mathcal{I}_i} 1}, \quad i \in \{1, \ldots, N_v\}, \tag{3}$$

where $\mathcal{I}_i$ is the index set of points within $i$-th voxel. Next, to enrich the voxel with geometric-semantic context, we project each voxel center into the input views and sample the corresponding features from $\mathbf{F}_t$, which are then concatenated with the initial voxel feature $\mathbf{v}_i^{\text{init}}$. The resulting 3D scaffold $\mathbf{S}_t$ of the scene is formally defined as a set of these voxels:

$$\mathbf{S}_t = \{(\mathbf{v}_i \in \mathbb{R}^{C_s}, \mathbf{p}_i \in \mathbb{R}^3)\}_{i=1}^{N_v} \tag{4}$$

where $C_s$ is the feature dimension, $\mathbf{v}_i$ represents the voxel feature encoding both geometric and semantic context, and $\mathbf{p}_i$ denotes the corresponding voxel center that preserves explicit 3D structure.

### 3.3 Unified Spatio-Temporal Scaffold Fusion

A key advantage of our scaffold representation lies in its inherent structure, which encodes explicit 3D geometry within a unified ego-centric space. This design enables contextual interaction in the unified 3D space, supporting direct and efficient spatio-temporal fusion across multiple views and temporal frames within a single scaffold representation.

**Spatial Scaffold Fusion.** Unlike traditional approaches Chen et al. (2024); Xu et al. (2025) that fuse spatial information across views in 2D space using image-level cross-attention, which is often hindered by limited overlap between views, we perform spatial fusion directly in the 3D scaffold space. In this representation, spatially corresponding information from different views is naturally aligned in 3D space. Specifically, we employ a sparse 3D U-Net $\phi$ to integrate multi-view features, producing a spatially-enhanced scaffold representation $\mathbf{S}_t^{\text{spa}}$:

$$\mathbf{S}_t^{\text{spa}} = \phi(\mathbf{S}_t), \tag{5}$$

**Temporal Scaffold Fusion.** Instead of processing historical raw images as in existing works (Lu et al., 2024; Tian et al., 2025), we integrate temporal cues directly within the scaffold representation in a streaming manner. Given the previous fused latent scaffold features $\mathbf{S}_{t-1}^{\text{fused}}$ from a streaming memory, we first warp its voxel centers into the current frame's coordinate system using the known ego-pose $T_{t-1}^t$, and their features are tagged with a time-step embedding to distinguish them from current observations. We then merge the transformed previous scaffold $\mathbf{S}_{t-1}^{\text{fused}}$ with the current scaffold $\mathbf{S}_t^{\text{spa}}$ via element-wise addition at any overlapping voxels, and simply union the features for non-overlapping regions. We denote this operation as a sparse tensor addition:

$$\mathbf{S}_t^{\text{fused}} = \mathbf{S}_t^{\text{spa}} \oplus \text{Warp}(\mathbf{S}_{t-1}^{\text{fused}}, T_{t-1}^t) \tag{6}$$

where $\oplus$ denotes sparse tensor addition that aggregates features at overlapping voxel locations while preserving non-overlapping features from both sparse tensors. The resulting tensor $\mathbf{S}_t^{\text{fused}}$ is further refined by a lightweight sparse convolutional network to capture complex temporal dependencies and is cached back into the streaming memory to maintain long-term temporal information.

### 3.4 Dynamic-aware Gaussian Generation

Building upon the spatio-temporally fused scaffold $\mathbf{S}_t^{\text{fused}}$, we generate a set of 3D Gaussian primitives via a dual-branch decoding strategy, yielding primitives that explicitly disentangle static and dynamic scene components, which enables progressive scene completion over time.

**Dual-Branch Gaussian Decoder.** Our Gaussian decoder comprises two complementary branches that jointly enhance reconstruction fidelity and completeness. The point decoder branch focuses

on preserving fine-grained geometric details by leveraging the point-level anchors from the reconstructed metric-scale point map $\mathbf{P}_t$. For each point $\mathbf{P}_{t,i} \in \mathbf{P}_t$, we locate its voxel coordinate in the scaffold and retrieve the corresponding latent feature from $\mathbf{S}_t^{\text{fused}}$ as:

$$f_{t,i}^{\text{3d}} = \text{Retrieve}\left(\mathbf{S}_t^{\text{fused}}, \left\lfloor \frac{\mathbf{P}_{t,i} - \mathbf{p}_{\min}}{\epsilon} \right\rfloor\right), \tag{7}$$

where $\lfloor \cdot \rfloor$ denotes the voxel indexing operation. If a point falls outside the scaffold's spatial extent, zero-padding is applied. Since each point $\mathbf{P}_{t,i}$ maintains a one-to-one correspondence with its source pixel, we additionally sample 2D image feature $f_{t,i}^{\text{2d}}$ for each point from the multi-view feature maps $\mathbf{F}_t$. These features are concatenated to predict the Gaussian primitives via an MLP:

$$\{(\Delta\boldsymbol{\mu}_i, \alpha_i, \boldsymbol{\Sigma}_i, \mathbf{c}_i, d_i)\} = \text{MLP}([f_{t,i}^{\text{3d}}, f_{t,i}^{\text{2d}}]), \tag{8}$$

where $\Delta\boldsymbol{\mu}_i$ denotes the Gaussian's offset from the point anchor, and $d_i \in \mathbb{R}$ is a learned dynamic score indicating motion likelihood. This branch yields a detailed set of Gaussians denoted as $\mathcal{G}_t^{\text{point}}$.

The voxel decoder branch complements the point-based decoding by directly predicting new Gaussian primitives from voxel-level scaffold features, effectively filling in sparsely covered regions and enhancing the scene completeness. For each voxel in $\mathbf{S}_t^{\text{fused}}$, we adopt a compact MLP to produce $g$ sets of Gaussian parameters (as in Eq. 8) per voxel. The center of each Gaussian is derived by adding the predicted displacement to the voxel center, forming the set $\mathcal{G}_t^{\text{voxel}}$. The complete reconstruction at time $t$ is then given by $\mathcal{G}_t = \mathcal{G}_t^{\text{point}} \cup \mathcal{G}_t^{\text{voxel}}$.

**Dynamic-aware Gaussian Completion.** To enhance temporal consistency and alleviate occlusion-induced sparsity, we introduce a memory mechanism that maintains accumulated static Gaussians over time. Each Gaussian primitive is associated with a dynamic attribute $d_i$, enabling motion-aware filtering. Given the static memory $\mathcal{M}_{t-1}$ from the previous frame, we transform it into the current ego-centric coordinate system and perform a view filtering to remove Gaussians visible in the current field of view. The resulting filtered memory $\mathcal{M}'_{t-1}$ is then fused with the current reconstruction:

$$\mathcal{G}_t^{\text{complete}} = \mathcal{G}_t \cup \mathcal{M}'_{t-1} \tag{9}$$

where $\mathcal{G}_t^{\text{complete}}$ provides a comprehensive scene representation that fills in the blind spots of the current frame's reconstruction. Finally, the memory is updated by retaining static Gaussians from the current frame:

$$\mathcal{M}_t = \mathcal{M}'_{t-1} \cup \{G_i \in \mathcal{G}_t \mid d_i < \tau_d\}, \ \ i \in \{1, ..., N_{\mathcal{G}_t}\} \tag{10}$$

where $\tau_d$ is a score threshold, and $N_{\mathcal{G}_t}$ is the total number of current Gaussians. This streaming mechanism enables temporally persistent reconstruction while suppressing dynamic artifacts.

## 3.5 TRAINING OBJECTIVE

The model is optimized via a composite loss function defined over the rendered outputs from $\mathcal{G}_t$:

$$\mathcal{L} = \sum_{v \in \mathcal{V}_{\text{input}}} \left(\lambda_1 \mathcal{L}_{\text{mse}}^v + \lambda_2 \mathcal{L}_{\text{lpips}}^v + \lambda_3 \mathcal{L}_{\text{dyn}}^v + \lambda_4 \mathcal{L}_{\text{scale}}^v\right) + \sum_{v \in \mathcal{V}_{\text{novel}}} \lambda_1 \mathcal{L}_{\text{mse}}^v \odot B^v \tag{11}$$

where $\mathcal{L}_{\text{mse}}^v$ and $\mathcal{L}_{\text{lpips}}^v$ are the MSE reconstruction and LPIPS perceptual losses (Zhang et al., 2018) between rendered and ground-truth images for view $v$, $\mathcal{L}_{\text{dyn}}^v$ is the cross-entropy loss between rendered dynamic scores and ground-truth dynamic segmentation masks, and $\mathcal{L}_{\text{scale}}^v$ is a smooth-L1 loss for scale supervision. $\mathcal{V}_{\text{input}}$ refers to the set of input camera views at time $t$ and $\mathcal{V}_{\text{novel}}$ denotes novel viewpoints at time $t + 1$. The operator $\odot$ denotes element-wise multiplication, where the background mask $B^v$ excludes dynamic regions to prevent optimization instability. Further details regarding dynamic rendering are provided in Appendix A.1.

## 4 EXPERIMENTS

### 4.1 EXPERIMENTAL SETTINGS

**Datasets and Metrics.** We conduct experiments on two large-scale autonomous driving benchmarks: Waymo Open (Sun et al., 2020) and nuScenes (Caesar et al., 2020) datasets. The Waymo

Table 1: Quantitative results on the Waymo Dataset. The best results are marked in **bold** and underlined entries indicate second-place performance. *: Evaluation conducted on front 3 views only. †: Results obtained using optimal scale alignment.

| Method | Views | Reconstruction | | | Novel View Synthesis | | |
|---|---|---|---|---|---|---|---|
| | | PSNR↑ | SSIM↑ | LPIPS↓ | PSNR↑ | SSIM↑ | LPIPS↓ |
| EvolSplat (Miao et al., 2025) | Front | 23.35 | 0.70 | 0.29 | - | - | - |
| UniSplat | Front | **28.93** | **0.86** | **0.18** | **27.34** | **0.80** | **0.20** |
| DriveRecon* (Lu et al., 2024) | Multi | 23.86 | 0.72 | 0.33 | 17.32 | 0.58 | 0.53 |
| MVSplat (Chen et al., 2024) | Multi | 24.94 | 0.80 | 0.23 | 22.04 | 0.68 | 0.34 |
| DepthSplat (Xu et al., 2025) | Multi | 25.38 | 0.76 | 0.26 | 23.86 | 0.70 | 0.31 |
| UniSplat | Multi | **28.56** | **0.83** | **0.20** | **25.12** | **0.74** | **0.27** |
| UniSplat† | Multi | 29.58 | 0.86 | 0.17 | 25.98 | 0.76 | 0.24 |

Table 2: Quantitative results on the nuScenes Dataset. We highlight best results in **bold** and second-place results with underlines. *: reported by Wei et al. (2025).

| Method | PSNR↑ | SSIM↑ | LPIPS↓ |
|---|---|---|---|
| PixelSplat* (Charatan et al., 2024) | 21.51 | 0.616 | 0.372 |
| MVSplat* (Chen et al., 2024) | 21.61 | 0.658 | 0.295 |
| Omin-Scene (Wei et al., 2025) | 24.27 | 0.736 | **0.237** |
| UniSplat | **25.37** | **0.765** | 0.246 |

Open dataset includes 798 training and 202 validation sequences, with all sequences approximately 20 seconds long and captured at 10Hz using five cameras. For nuScenes, which provides six surround-view images per frame, we adopt the strategy of Wei et al. (2025) and partition scenes into equally spaced bins along the vehicle trajectory, yielding 135,941 training and 30,080 validation bins. Each bin consists of multiple sequential frames, and the central frame serves as the input.

To measure visual quality, we adopt standard image quality metrics including PSNR, SSIM (Wang et al., 2004), and LPIPS (Zhang et al., 2018). Following Yang et al. (2024); Lu et al. (2024), the Waymo benchmark evaluates two tasks: reconstruction, for which images at current timestep $t$ serve as targets, and novel view synthesis, which synthesizes images at the subsequent timestep $t+1$. For nuScenes, consistent with Wei et al. (2025), we evaluate on target views consisting of the first, last, and central frames of each bin.

**Implementation Details.** For our 3D scaffold reconstruction, we employ a frozen pretrained geometry transformer $\pi^3$ (Wang et al., 2025f) for initial geometry generation and a pretrained DINOv2 ViT-small backbone (Oquab et al., 2023) for semantic feature extraction. The scaffold is built within a real-world volume of [-72m, -72m, -4m, 72m, 72m, 12m], using an initial voxel size of (0.1m, 0.1m, 0.2m). Scaffold spatial fusion is performed using a sparse 3D U-Net with a maximum downsampling factor of $8\times$, while the temporal fusion employs a separate sparse 3D U-Net with a maximum downsampling factor of $2\times$. In the Gaussian decoding stage, the second branch generates $g=4$ primitives per voxel, and the dynamic attribute threshold for streaming scene completion is set to $\tau_d = 0.7$. We adopt image resolutions of $350 \times 518$ for the Waymo dataset and $224 \times 406$ for the nuScenes dataset. All models are trained using the AdamW optimizer (Loshchilov & Hutter, 2019) on 16 NVIDIA H20 GPUs with a total batch size of 32. For the training objective, we set $\lambda_1$=1.0, $\lambda_2$=0.01, $\lambda_3$=0.01, and $\lambda_4$=0.02. Additional implementation details are provided in Appendix A.1.

## 4.2 MAIN RESULTS

**Waymo.** We compare UniSplat against state-of-the-art sparse-view reconstruction methods, including MVSplat (Chen et al., 2024), DepthSplat (Xu et al., 2025), EvolSplat (Miao et al., 2025), and DriveRecon (Lu et al., 2024). For the general methods MVSplat and DepthSplat, we retrain them on the Waymo Open Dataset using their official codebases. For driving-specific methods EvolSplat and DriveRecon, we conduct evaluation on our validation scenes and resize their outputs to match the resolution for fair comparison. The quantitative results are summarized in Table 1. UniSplat consistently outperforms all baselines across every metric for both input view reconstruction and novel view synthesis. The qualitative comparisons are shown in Figure 2. Notably, MVSplat and DepthSplat struggle to reconstruct fine geometric details and exhibit noticeable artifacts, especially in overlapping regions between adjacent cameras. In contrast, our method produces visually coher-

ent and high-quality results. We also report an variant (denoted by †), in which per-camera scales are set to optimal values derived from LiDAR pointmap, leading to additional improvements.

**NuScenes.** Following Wei et al. (2025), we evaluate UniSplat on the nuScenes benchmark under the same protocol. As shown in Table 2, UniSplat surpasses the previous state of the art, Omni-Scene, achieving 25.37 PSNR (+1.10dB). The qualitative comparisons are provided in Appendix A.3.

**Dynamic-aware Gaussian Completion.** UniSplat predicts per-Gaussian dynamic attributes, enabling the progressive construction of the scene during inference without manual labels. As shown in Figure 3, the top section presents a baseline without dynamic filtering, where ghosting artifacts arise from accumulated dynamic objects. In contrast, our approach effectively completes missing regions while suppressing such artifacts. As illustrated in the bottom section, UniSplat successfully completes unobserved areas arising from two typical cases: limited 360° coverage in Waymo's five-camera setup and cross-camera blind spots. Moreover, we can observe our model clearly separates dynamic vehicles from parked ones, demonstrating its effective use of temporal context.

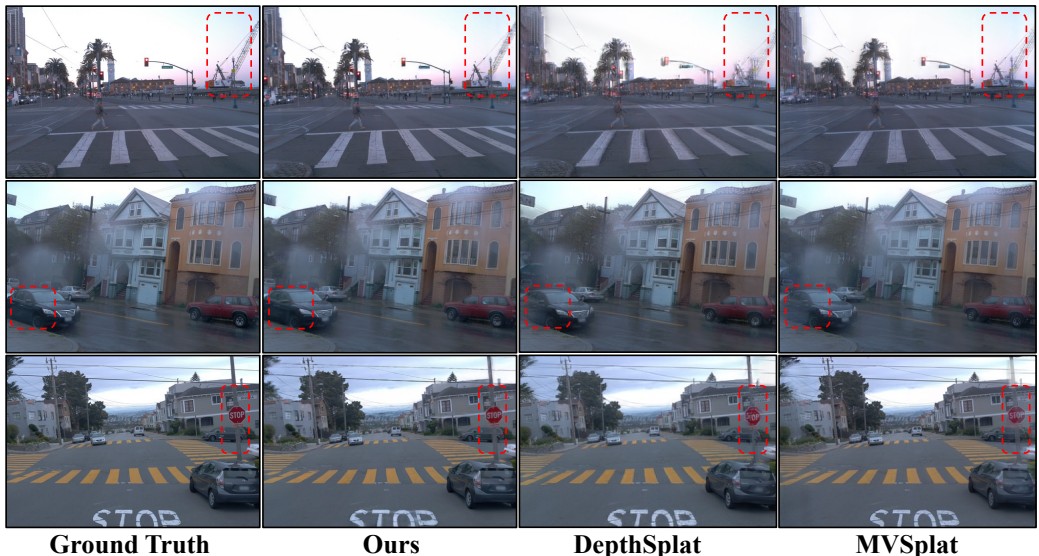

| Ground Truth | Ours | DepthSplat | MVSplat |

Figure 2: Qualitative comparisons on the Waymo dataset. Our method yields more detailed and consistent geometry than existing works. Red boxes indicate artifacts. Best viewed zoomed in.

### 4.3 ABLATION STUDY

In this section, we conduct ablation studies on the Waymo Open Dataset (Sun et al., 2020) to investigate the individual components of our framework, with a focus on novel view synthesis performance. For efficiency, we subsample the first 20% of frames from each sequence and apply optimal scale alignment to the point map to accelerate model convergence. All models are trained for 20 epochs with a batch size of 32 on 16 GPUs.

**Ablation on Geometric and Semantic Features in Scaffold.** Table 3 investigates the contribution of geometric and semantic features from foundation models to the scaffold representation. The absence of semantic features causes a severe decline in LPIPS, increasing the error by 0.05, which can be attributed to the fact that LPIPS measures perceptual similarity using deep semantic representations. In contrast, the $2^{nd}$ and $3^{rd}$ rows show that performance gap is less pronounced when only DINO features are used, suggesting that current large-scale pretrained 2D foundation model (Siméoni et al., 2025) may implicitly capture certain geometric priors.

**Analysis of Spatio-Temporal Fusion.** We ablate the effects of our spatial and temporal scaffold fusion, with results summarized in Table 4. As shown in $1^{st}$ and $2^{nd}$ rows, the incorporation of spatial scaffold fusion, which aggregates spatial information in 3D space, improves performance by +0.36dB in PSNR and +0.02 in SSIM compared to the baseline that only relies on image-domain fusion. Further integration of temporal scaffold fusion, which incorporates historical context through ego-motion warping and fusion in the latent scaffold domain, brings an additional gain of +0.58dB

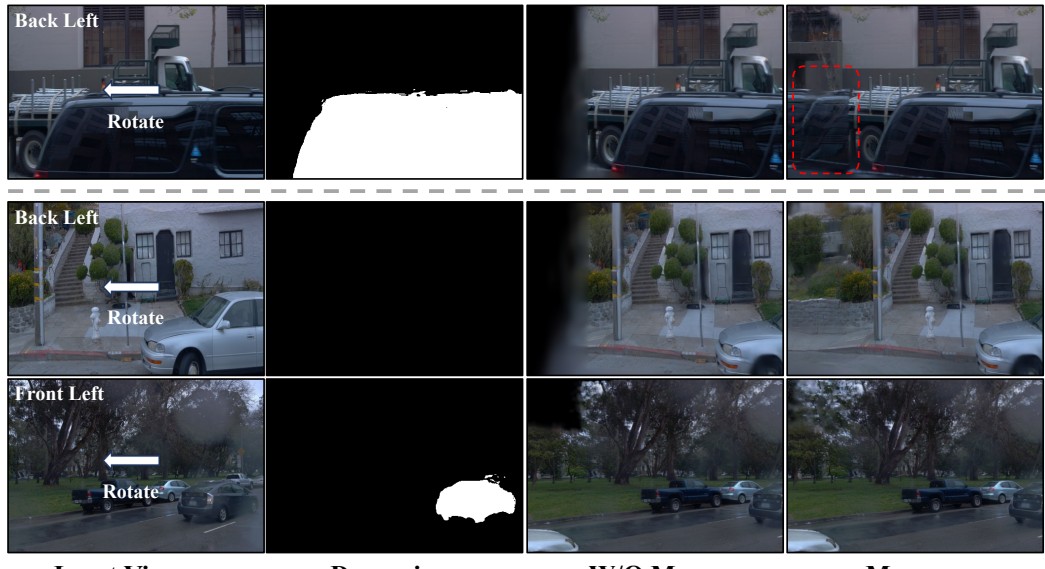

Figure 3: Qualitative results of scene completion on the Waymo dataset. **Top**: Aggregated scene without dynamic filtering, where red boxes indicate ghosting artifacts caused by accumulating the dynamic car. **Bottom**: Our method, equipped with dynamic-aware Gaussians, completes unobserved regions due to limited sensor coverage and bridges cross-camera gaps while avoiding dynamic artifacts. The predicted dynamic masks used for filtering are shown for reference.

Table 3: Impact of feature composition of $\mathbf{F}_t$. "Geo" and "Sem" denote geometric and semantic features, respectively.

| Geo | Sem | PSNR↑ | SSIM↑ | LPIPS↓ |
|-----|-----|-------|-------|--------|
| ✓ | | 24.78 | 0.73 | 0.35 |
| | ✓ | 24.85 | 0.72 | 0.31 |
| ✓ | ✓ | **25.08** | **0.74** | **0.30** |

Table 4: Analysis of spatio-temporal fusion. "Spa" and "Tem" denote spatial and temporal fusion, respectively.

| Spa | Tem | PSNR↑ | SSIM↑ | LPIPS↓ |
|-----|-----|-------|-------|--------|
| | | 24.14 | 0.68 | 0.32 |
| ✓ | | 24.50 | 0.70 | 0.32 |
| ✓ | ✓ | **25.08** | **0.74** | **0.30** |

in PSNR and +0.04 in SSIM. We also compare against a variant that explicitly uses two consecutive frames without latent-space temporal propagation. This approach achieves a lower PSNR of 24.72dB, likely due to its limited ability to model dynamic elements and restricted temporal context. These results demonstrate the effectiveness of our unified spatio-temporal modeling approach that operates directly within the 3D scaffold representation for handling sparse, minimally-overlapping camera views and complex dynamic driving scenes.

**Dual-Branch Gaussian Decoder.** We validate our dual-branch decoder design in Table 5. Using only point-anchored Gaussians results in a performance degradation of 0.46 in PSNR, 0.02 in SSIM, and an increase of 0.08 in LPIPS error, underscoring the critical role of voxel-generated Gaussians in improving scene completeness by effectively filling sparsely covered regions. The voxel-only variant is excluded from comparison as it fails catastrophically at long-range rendering (Wei et al., 2025), yielding consistently poor performance across all metrics.

**Geometry Foundation Model.** In Table 6, We ablate the impact of the geometry foundation model on our framework's performance. Specifically, replacing the default model with MoGe-2 (Wang et al., 2025e), a recently introduced open-domain geometry estimation method, yields consistent performance, which indicates that our approach is robust to the choice of the underlying geometry foundation model. Notably, we exclude the representative VGGT (Wang et al., 2025a), as our empirical observations indicate that it generalizes less effectively than $\pi^3$ in outdoor driving scenarios.

## 5    CONCLUSION

We presented **UniSplat**, a unified feed-forward framework for dynamic driving scene reconstruction and novel view synthesis. Our core contribution is the introduction of a 3D latent scaffold

Table 5: Ablation study on the two branches of our Gaussian decoder.

| Point | Voxel | PSNR↑ | SSIM↑ | LPIPS↓ |
|---|---|---|---|---|
| ✓ | | 24.62 | 0.72 | 0.38 |
| ✓ | ✓ | **25.08** | **0.74** | **0.30** |

Table 6: Performance comparison of different geometry foundation models.

| Models | PSNR↑ | SSIM↑ | LPIPS↓ |
|---|---|---|---|
| MoGe-2 | 24.98 | 0.74 | **0.29** |
| $\pi^3$ | **25.08** | **0.74** | 0.30 |

that seamlessly unifies spatio-temporal fusion from multi-camera videos. By leveraging foundation models, this scaffold encodes robust geometric and semantic priors, enabling efficient fusion directly in 3D space. We further proposed a dual-branch Gaussian decoder that generates dynamic-aware primitives from the scaffold, coupled with a streaming memory mechanism to accumulate static scene content over time for long-term completion. Extensive experiments on Waymo and nuScenes demonstrate that UniSplat not only achieves state-of-the-art performance under standard settings but also exhibits remarkable generalization to challenging viewpoints outside the original camera coverage. We believe that our framework provides a promising foundation for future research on dynamic scene understanding, interactive 4D content creation, and lifelong world modeling.

## ACKNOWLEDGMENTS

This work is supported by Guangdong Basic and Applied Basic Research Foundation (2025A1515011434), Shenzhen Science and Technology Program (ZDCY20250901113000001), and CCF-DiDi GAIA Collaborative Research Funds.

## ETHICS STATEMENT

We confirm adherence to the ICLR Code of Ethics and have carefully evaluated the ethical implications of our research. Our work advances 3D reconstruction for autonomous driving, aiming to improve safety while encouraging responsible deployment. We utilize established public datasets (Waymo Open Dataset and nuScenes) in strict compliance with their licenses, without processing sensitive personal identifiers.

## REPRODUCIBILITY STATEMENT

To ensure reproducibility of our results, we have provided comprehensive details necessary to replicate our experiments and publicly released our source code. The main text outlines our experimental settings in Section 4.1, including dataset usage, evaluation metrics, and training configurations. Further implementation specifics are documented in Appendix A.1, which covers network architecture details, hyperparameter settings, and the use of software libraries such as SpConv for sparse convolutions. All experiments are based on publicly available datasets, including the Waymo Open Dataset and the nuScenes dataset, and use clearly defined data splits and evaluation protocols consistent with prior work. For a fair comparison with baseline methods, we describe the retraining procedures and adaptations in Appendix A.1.

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

# A   APPENDIX

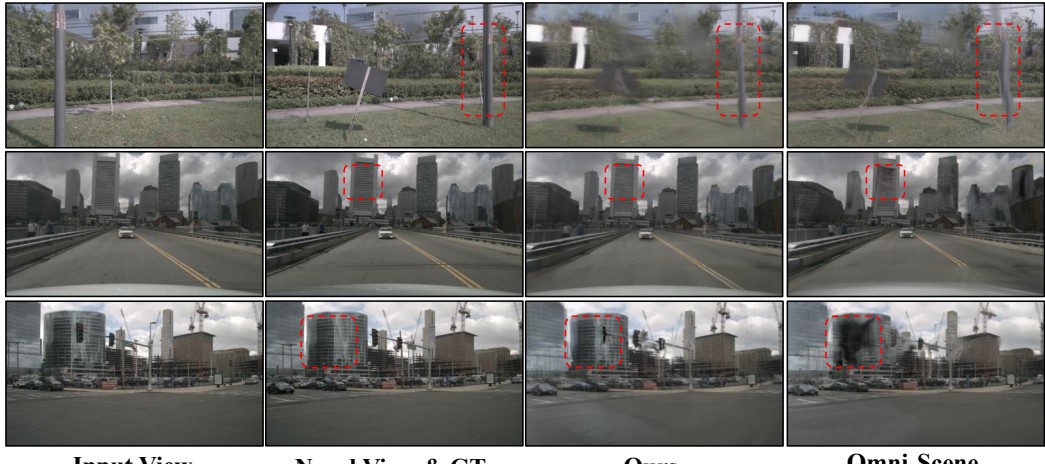

| Input View | Novel View & GT | Ours | Omni-Scene |

Figure 4: Qualitative comparisons on the nuScenes dataset. The red boxes highlight undesirable artifacts

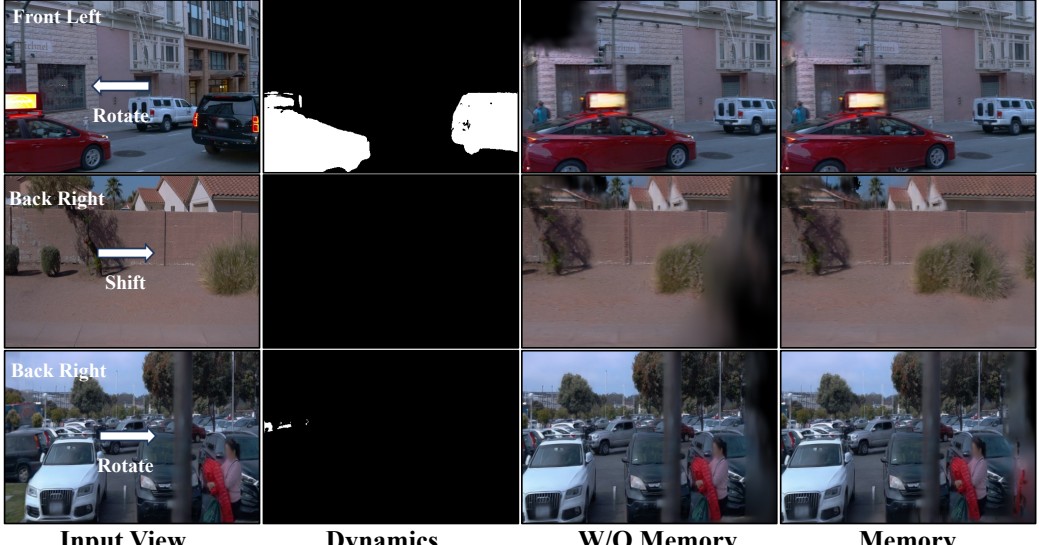

| Input View | Dynamics | W/O Memory | Memory |

Figure 5: Additional qualitative results of streaming scene completion on the Waymo dataset

## A.1   IMPLEMENTATION DETAILS

**Implementation details of UniSplat.** We employ the SpConv (Contributors, 2022) library to implement the sparse 3D U-Net, which comprises convolutional and transposed convolutional layers and achieves a maximum downsampling factor of $8\times$. The model is trained in a streaming manner using clips of 20 frames for 20 epochs, with an initial learning rate of $1.5 \times 10^{-4}$ following a cosine decay schedule. For the semantic backbone within the 3D scaffold reconstruction, we uses a learning rate scaled by a factor of 0.1. To address the severe class imbalance in the dynamic segmentation loss, we incorporate a negative sampling strategy that randomly selects 50,000 negative pixels per sample for loss computation. For Gaussian rasterization, we adopt the framework of Kerbl et al. (2023) and, following StreetGaussian (Yan et al., 2024), set the spherical harmonics (SH) degree to 1 for efficiency.

**Dynamic Rendering.** To supervise the dynamic attributes of the Gaussians in $\mathcal{G}_t$, we introduce a dynamics rendering mechanism that renders dynamic masks using the standard differentiable

Table 7: Efficiency comparison on the nuScenes dataset.

| Method | FPS↑ | Mem.(GB)↓ | Param(M) | PSNR↑ | SSIM↑ | LPIPS↓ |
|---|---|---|---|---|---|---|
| Omin-Scene (Wei et al., 2025) | 2.5 | **8.22** | 81.7 | 24.27 | 0.736 | **0.237** |
| UniSplat | **4.0** | 8.30 | 91.0 | **25.37** | **0.765** | 0.246 |

Gaussian-splatting pipeline, with dynamic logits as inputs instead of colors:

$$D = \sum_{i \in \mathcal{N}} d_i \alpha_i \prod_{j=1}^{i-1} (1 - \alpha_j), \tag{12}$$

Where $D$ denotes the per-pixel dynamic probability. For ground-truth mask generation, we identify moving objects via 3D bounding box tracking, project them onto the image plane to create prompts for SAM2 (Ravi et al., 2024), and subsequently use the model to generate the final masks.

**Implementation details of UniSplat counterparts.** To adapt general feedforward reconstruction baselines to the autonomous driving setting, we retrain MVSplat (Chen et al., 2024) and Depth-Splat (Xu et al., 2025) on the Waymo Open Dataset Sun et al. (2020). For MVSplat, we initialize the model using its official weights pre-trained on RealEstate10K (Zhou et al., 2018). Context views are from the current timestep, while target viewpoints include both the current and next timesteps. Training is conducted with a batch size of 16 on 8 H20 GPUs for 40,000 iterations, as further training empirically degrades performance. For DepthSplat, we initialize from its official weights pre-trained on dl3dV (Ling et al., 2024) and use the variant equipped with a ViT-B backbone (Dosovitskiy et al., 2021). All other training settings remain consistent with those used for MVSplat.

**Evaluation Protocol on NuScenes.** Due to the patch size constraint of our geometry foundation model, which requires image dimensions to be divisible by 14, we train our model at a resolution of $224 \times 406$, differing from the $224 \times 400$ resolution used by Omni-Scene (Wei et al., 2025). For a fair comparison, evaluation is performed by resizing our model's outputs to $224 \times 400$, aligning with the baseline's resolution before metric computation.

## A.2 EFFICIENCY ANALYSIS

We benchmark the efficiency of our method against Omni-Scene, a state-of-the-art open-source driving-specific reconstruction model, on the nuScenes dataset (Caesar et al., 2020). Note that Omni-Scene initializes its pixel-aligned Gaussians using predictions from a pretrained monocular depth estimation model (Hu et al., 2024), which are precomputed and not included in its computational cost. To ensure a fair comparison, we also exclude the cost of our geometry foundation model during inference. The results are summarized in Table 7. UniSplat attains higher runtime efficiency (4.0 FPS vs. 2.5 FPS) while surpassing Omni-Scene by a large margin in reconstruction quality. We attribute this to our fine-grained spatial fusion and streaming temporal aggregation in latent scaffold space. We also observe that Omni-Scene's rendering stage is the primary bottleneck (60% of inference time), as it generates roughly 2 million voxel-based Gaussians per scene. All experiments were conducted on a single H20 GPU. The reported inference time represents the end-to-end reconstruction and rendering of all 18 target frames per sample, averaged over 2,048 samples, with data loading time excluded.

## A.3 MORE QUALITATIVE RESULTS.

**Qualitative Comparisons on the nuScenes dataset.** Qualitative comparisons for novel view synthesis against Omni-Scene are presented in Figure 4. Our method demonstrates superior spatial coherence, as evidenced in challenging cases such as the thin pole (first row), and produces fewer artifacts like the buildings shown in the second and third rows.

**Streaming Scene Completion.** Figure 5 provides additional qualitative results for our streaming scene completion capability. As shown in the first and second rows, when the viewpoint rotates or shifts to the regions outside the camera frustums, our method robustly reconstructs these newly visible areas, maintaining high fidelity and spatial coherence. The third row illustrates a failure case in which a moving pedestrian is misclassified as static. As a result, the dynamic object is improperly retained in the memory, leading to noticeable ghosting artifacts in the rendered sequence.

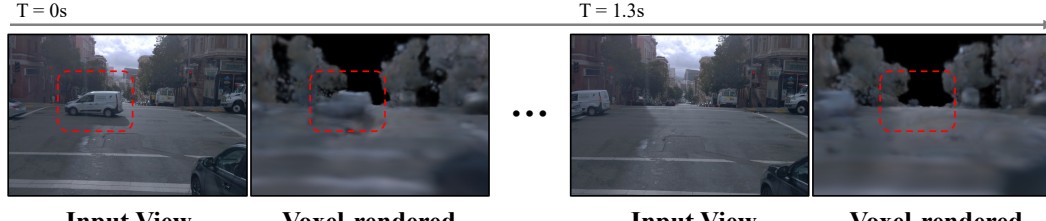

Figure 6: Visualization of a moving vehicle at two timestamps (T = 0 s and T = 1.3 s). For each time step, we show the input view and the corresponding voxel-rendered result from our scaffold. Despite the vehicle's motion, the renderings exhibit no ghosting artifacts or temporal inconsistencies.

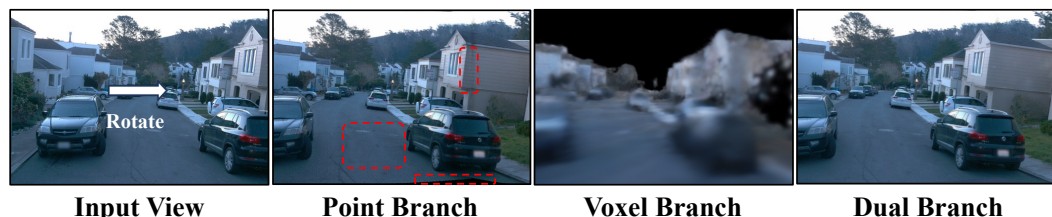

Figure 7: Reconstructions from the point-only, voxel-only, and dual-branch decoders under camera rotation. The red boxes highlight artifacts that appear when using only point branch.

**Dynamic Handling in the Latent Scaffold.** Beyond the explicit dynamic filtering employed in our streaming Gaussian memory, we observe that the latent scaffold itself exhibits an inherent ability to handle scene dynamics. Through unified spatio-temporal fusion, the model implicitly learns to aggregate multi-frame features according to geometric consistency. In Figure 6, we visualize the reconstruction of a scene containing a moving vehicle. Although features from the moving vehicle are repeatedly integrated into nearby scaffold voxels over time, the voxel-rendered results remain free of trailing artifacts or temporal inconsistencies. This suggests that the learned fusion in the latent space effectively integrates temporal information and suppresses outdated evidence from dynamic objects.

**Visual Analysis of the Dual-Branch Decoder.** To better illustrate the behavior of the point and voxel branches, we visualize the rendering outputs from individual branches under a large view-point change in Figure 7. The point branch preserves high-frequency details but leads to overfitting to the input view, resulting in holes and distortions in the novel view (highlighted in red). Conversely, the voxel branch serves as a continuous volumetric backbone, although it tends to produce smoother reconstructions with limited fine-grained sharpness. The final dual-branch decoder effectively combines these complementary strengths, recovering sharp details while maintaining robust structural integrity in novel views.

## A.4 DYNAMIC ACTOR EDITING

Leveraging the explicit disentanglement of static and dynamic Gaussians, UniSplat supports flexible scene manipulation tasks, including the removal, relocation, and insertion of dynamic actors.

**Extracting dynamic actors.** Using the learned dynamic scores, we first render a 2D dynamic mask and extract connected components. 3D Gaussians corresponding to the selected region with high dynamic probabilities are then grouped as independent actor assets.

**Background restoration after editing.** A primary challenge in object removal or relocation is the "disocclusion" problem, where the background region behind a moving object is unobserved in the current frame. Our framework addresses this by leveraging the streaming memory. In the standard pipeline, we employ a view-filtered memory $\mathcal{M}'_{t-1}$ (Eqs. 9 and 10) to avoid redundancy with the current observations. For editing tasks, however, we explicitly query the full memory state $\mathcal{M}_{t-1}$, which allows us to recover static Gaussians captured at earlier timestamps but occluded by the dynamic actor in the current view.

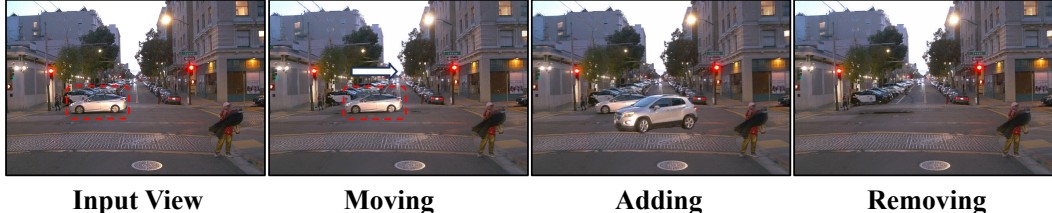

| **Input View** | **Moving** | **Adding** | **Removing** |

Figure 8: Dynamic actor editing. Starting from the **Input View** (left), we demonstrate three editing operations: **Moving** the vehicle, **Adding** a vehicle instance to the scene, and **Removing** the vehicle entirely.

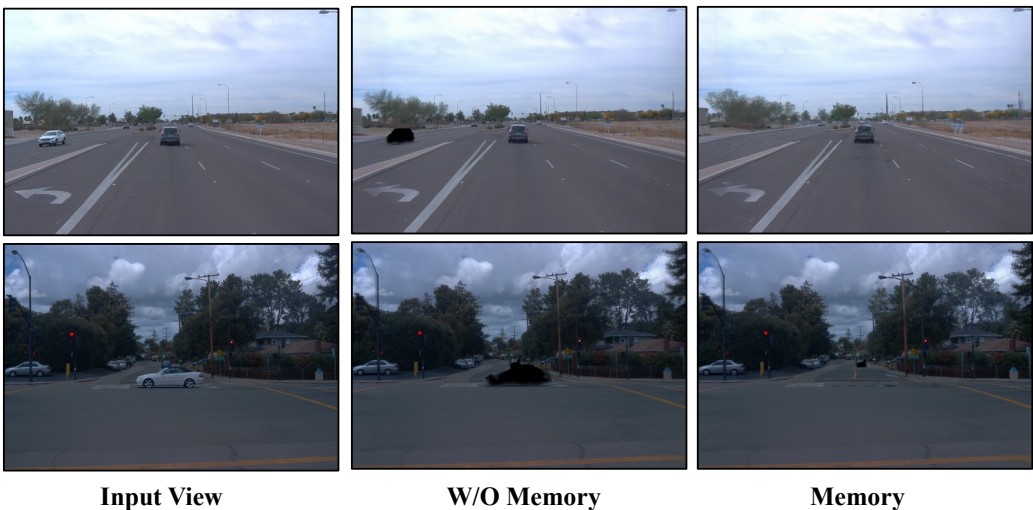

| **Input View** | **W/O Memory** | **Memory** |

Figure 9: More qualitative results of actor removing.

**Unified re-rendering with edited actors.** Finally, the edited dynamic Gaussians (e.g., moved to a new position or newly inserted) are combined with the restored static scene. This unified set is rendered to produce the manipulated scene. As illustrated in Figure 8 and Figure 9, we demonstrate successful manipulation operations, including moving, adding, and removing an actor, highlighting the geometric consistency of both the edited actor and the recovered background.

## A.5 VIDEO VISUALIZATION

We present a supplementary video demonstrating our reconstruction results on two scenes, showcasing novel view synthesis under camera shifts, along with dynamic prediction capabilities. We note that in certain frames (e.g., at 1:03), the low Gaussian opacity of glass surfaces results in a perceived misalignment between dynamic masks and RGB content. Please refer to the video file included in the supplementary material.

## B DECLARATION OF LLM USAGE

Large Language Models (LLMs) were utilized to assist with language refinement and manuscript preparation, including grammar checking and enhancing textual clarity. All scientific concepts, methodological innovations, experimental frameworks, data analysis, and conclusions presented in this work are independently developed by the authors. We have thoroughly reviewed and validated all content, and assume complete responsibility for the accuracy and integrity of this manuscript.

