# OpenReview forum: "UniSplat: Unified Spatio-Temporal Fusion via 3D Latent Scaffolds for Dynamic Driving Scene Reconstruction"
_ICLR.cc/2026/Conference — ICLR 2026 Poster_

### Official Review · Reviewer_2VZv · 2025-10-20

**Soundness:** 3
**Presentation:** 3
**Contribution:** 2
**Rating:** 6
**Confidence:** 3

**Summary:**

This paper proposes UniSplat, a feed-forward framework for dynamic driving scene reconstruction based on 3D Gaussian Splatting, introducing a 3D latent scaffold that enables unified spatio-temporal feature fusion across multi-view camera inputs and sequential frames. A dual-branch Gaussian decoder jointly handles fine-grained reconstruction and spatial completion, while a streaming Gaussian memory with dynamic-aware filtering progressively accumulates static scene content to address coverage gaps beyond the current field of view.

**Strengths:**

1. The 3D latent scaffold and spatio-temporal fusion mechanism introduce a more structured way to aggregate multi-view and temporal cues, beyond 2D feature aggregation used in previous feed-forward approaches.
2. Dynamic-aware Gaussian memory is conceptually appealing, enabling persistent scene completion and reducing ghosting artifacts from moving objects.

**Weaknesses:**

1. Readers may find it difficult to understand why the scaffold representation is fundamentally more suitable for handling sparse multi-view driving scenarios.
2. The streaming Gaussian memory mechanism is effective in mitigating ghosting and compensating camera blind spots. However, it is introduced as a component-level implementation rather than a more formalized recurrent world-state update.
3. The dual-branch decoder (point-based detail refinement and voxel-based completion) shows quantitative gains, but the ablations do not analyze where each branch contributes most (e.g., near-field vs.\ far-field geometry, static vs.\ dynamic regions).

**Questions:**

1. Regarding the streaming memory, can the authors provide more insight into how the memory size evolves over long sequences? Does the model include any mechanism for memory pruning or confidence-based discarding, and could memory saturation impact performance?
2. For the dual-branch decoder, have the authors observed complementary behaviors between point-anchored and voxel-generated Gaussians across different scene structures? A visual or statistic showing their relative contributions would help clarify the motivation.
3. How sensitive is the overall system to the accuracy of the geometry foundation model used for initializing the scaffold? For example, if the model produces slight drift or inconsistent scale, could this propagate and affect memory fusion stability?

---

> ### Author Response · Authors · 2025-11-21
> **Response to Reviewer 2VZv (1/2)**
>
> **We sincerely thank you for your valuable suggestions. Here are our responses to your questions:**
>
> >**W1. Advantages of the 3D scaffold for sparse multi-view driving scenarios:**
>
> We appreciate the reviewer’s concern, and here we clarify the fundamental advantages of our 3D latent scaffold:
>
> **Spatial fusion.** In autonomous driving, the overlap between adjacent cameras is often minimal. Purely 2D fusion (e.g., cross-attention) struggles in this regime as it relies on visual correspondence across views. In contrast, our scaffold lifts features into a unified 3D metric space, where spatially corresponding information from different views is naturally aligned, regardless of visual overlap. This allows our model to effectively fuse context and complete geometry in regions where 2D attention would fail.
>
> **Temporal fusion.** Operating in an ego-centric 3D scaffold also stabilizes temporal fusion. The scaffold from the previous frame can be explicitly warped to the current timestep using known poses, which naturally disentangles ego-motion from object dynamics. Such disentanglement is inherently difficult in the 2D image domain, where both motions are coupled. Furthermore, this streaming memory also provides a robust long-horizon spatio-temporal context for reconstruction.
>
>
> >**W2. About streaming Gaussian memory:**
>
> We acknowledge that our streaming Gaussian memory is implemented as a component-level mechanism, as our primary goal is to efficiently compensate camera blind spots and extend static scene coverage. Conceptually, however, our framework already incorporates a recurrent world-state update within the Latent Scaffold. The historical scaffold is warped and fused with current observations, while these observations in turn update the scaffold. We agree that casting the Gaussian memory updates within a more formal probabilistic or recurrent framework is a promising direction, but we consider it complementary to our current efficient design.
>
> >**W3. Analysis of the dual-branch decoder:**
>
> We designed the dual-branch decoder to exploit the complementary strengths of point-anchored and voxel-generated Gaussians. Based on our ablation studies and visual analysis, we observe distinct roles for each branch across different scene structures:
>
> **Point-anchored branch.** This branch excels at recovering high-frequency details and sharp textures due to dense point inputs from the geometry prior. However, relying solely on points often leads to overfitting to the input views, which tends to produce rendering artifacts when the viewpoint changes.
>
> **Voxel-generated branch.** This branch generates Gaussians from voxel centers, serving as a coarse skeleton of the scene. It contributes most in textureless regions (e.g., road surfaces) and geometric blind spots where point estimation is sparse or unreliable. While voxel-only rendering is generally less sharp and may miss fine details, it provides better topological completeness.
>
> As shown in **Figure 7 (Appendix Sec A.3, page 17),** we visualize the rendering results under a 12-degree rotation from the input view. The "Point Branch" result exhibits structural discontinuities (e.g., cracks in road surfaces) and empty holes in unseen regions, whereas the "Dual Branch" result maintains a coherent and complete scene structure.

---

> > ### Author Response · Authors · 2025-11-21
> > **Response to Reviewer 2VZv (2/2)**
> >
> > >**Q1. Evolution of memory size and pruning mechanisms:**
> >
> > Our streaming memory is explicitly designed to remain bounded and robust over long sequences. We manage the memory at two levels: the latent scaffold memory (feature level) and the static Gaussians memory (primitive level).
> > **Scaffold memory.** On Waymo sequences (~20 s, 200 frames), our 3D scaffold construction and spatio-temporal fusion run at 80–100 ms per frame, and GPU memory remains stable at ~16–17 GB over the full sequence. This bounded behavior comes from two designs:
> >
> > * **Spatial range constraint.** The scaffold is defined only within a fixed spatial region around the current ego pose. Historical voxels that drift outside this window are automatically discarded during fusion.
> >
> > * **Adaptive pruning.** Before fusion, we estimate a confidence score for each voxel in the historical scaffold by averaging the opacities of the Gaussians generated from its corresponding voxel branch. We retain only the top 60% high-opacity voxels, effectively filtering out low-confidence historical content to keep the memory compact.
> >
> > **Static Gaussians memory.**  The number of stored Gaussians grows linearly with the spatial area explored. On Waymo, we observe an approximately linear growth, with about 100–300 thousand additional Gaussians per frame, with the exact value depending on ego-velocity. Importantly, we only keep Gaussians from regions outside the current FOV, thus avoiding near-duplicate Gaussians for the same visible content. For indefinitely long streams, our framework can be readily extended with distance-based culling or offloading to disk, offering a flexible trade-off between memory usage and the range of historical reconstruction.
> >
> > >**Q2. Complementary role of the dual-branch decoder:**
> >
> > Please refer to W2.
> >
> > >**Q3. Sensitivity to the geometry foundation model:**
> >
> > We acknowledge that geometry foundation models may exhibit scale inconsistency or drift. However, UniSplat demonstrates strong resilience to such initialization imperfections, particularly for our primary task of novel view synthesis. We perturb the foundation geometry $\mathbf{P}_t$ by adding zero-mean Gaussian noise to its global scale and shift, and measure novel-view PSNR on 10 randomly selected validation scenes:
> > | $\sigma_{\text{scale}}$ |  $\sigma_{\text{shift}}$ | PSNR (Novel View) | $\Delta$ PSNR |
> > | :---: | :---: | :---: | :---: |
> > | 0.00 | 0.00m | **25.99** | - |
> > | 0.02 | 0.10m | 25.97 | -0.02 |
> > | 0.04 | 0.20m | 25.92 | -0.07 |
> > | 0.08 | 0.40m | 25.74 | -0.25 |
> >
> > The degradation remains modest even under non-trivial perturbations, indicating limited sensitivity to initialization errors. This robustness arises because the geometry prior is used only as a coarse initialization, which is subsequently refined by the fusion module using accumulated multi-view and multi-frame evidence.
> >
> > While the system is locally robust, we acknowledge that our current design faces challenges in reconstructing entire street-scale scenes with perfect global alignment due to accumulated drift over extended sequences. We believe incorporating multi-frame supervision from historical viewpoints or global bundle adjustment to rectify such errors represents a promising direction, and we plan to explore it in future work.

---

> > > ### Comment · Area_Chair_zyAr · 2025-11-28
> > >
> > > Dear Reviewer Reviewer 2VZv ,
> > >
> > > The authors gave their rebuttals to your reviews. Could you please express your opinions?
> > >
> > > Many thanks.
> > >
> > > AC

---

> > > > ### Comment · Reviewer_2VZv · 2025-11-28
> > > >
> > > > Thanks for the reply. I’ll keep my rating.

---

### Official Review · Reviewer_rymU · 2025-10-27

**Soundness:** 3
**Presentation:** 3
**Contribution:** 3
**Rating:** 6
**Confidence:** 4

**Summary:**

This paper introduces UniSplat, a unified framework for spatio-temporal fusion and dynamic scene reconstruction.
The key idea is to construct a latent 3D scaffold where both spatial fusion (within a frame) and temporal fusion (across frames) are performed in the same structured voxel domain.
The previous fused scaffold is warped to the current frame using ego-pose and merged directly, avoiding redundant computation and maintaining geometric consistency.
A dual-branch Gaussian decoder combines point-based details with voxel-based coverage, while a dynamic filtering mechanism accumulates only static components in a memory buffer to enable out-of-FOV reconstruction.
Experiments on the Waymo and nuScenes datasets show that UniSplat significantly improves reconstruction accuracy and efficiency over prior works such as MVSplat, Omni-Scene, and DriveDreamer4D.

**Strengths:**

Proposes a unified spatio-temporal fusion paradigm in a single latent 3D scaffold, which is a conceptually cleaner and more efficient design than previous separate spatial and temporal modules.
The dual-branch Gaussian decoder effectively balances fine-grained details (point) and global completeness (voxel).
The dynamic filtering + memory streaming mechanism elegantly addresses out-of-FOV reconstruction and ghosting issues caused by dynamic objects.
Strong experimental results and ablations demonstrate the necessity and contribution of each module.
The framework is feed-forward and efficient, making it more practical than diffusion- or transformer-based alternatives.

**Weaknesses:**

While the framework is conceptually unified, many of its components (voxel scaffold, temporal warping, Gaussian splatting) are adapted from prior work.
The dynamic filtering relies on threshold-based heuristics; it is not clear how robust this is under complex motion or sensor noise.
The paper lacks a deeper comparison with recent diffusion-based or token-based reconstruction frameworks, which could strengthen the positioning of this method.
Some design choices (e.g., the specific form of the dual-branch decoder, scaffold resolution) are under-explained, and might be perceived as empirical rather than principled.

**Questions:**

Why did you choose to perform spatio-temporal fusion entirely in the latent 3D scaffold domain rather than using separate spatial and temporal modules (as in Omni-Scene)? What specific benefits (efficiency, stability, or accuracy) did you observe?

Could you elaborate on why the dual-branch decoder was designed in its current form? Were other fusion strategies (e.g., adaptive weighting between point and voxel) explored?

How robust is the dynamic filtering threshold under varying dynamic motion or sensor noise? Could adaptive strategies improve generalization?

Why not adopt token-based or diffusion-based representations for the scaffold? Is the explicit voxel representation crucial for efficiency or accuracy?

How does the method behave when the camera view coverage becomes extremely sparse (e.g., two front-facing cameras only)?

Are there scenarios where the streaming memory might accumulate false positives from misclassified dynamic objects, and how would the method handle this?

---

> ### Author Response · Authors · 2025-11-21
> **Response to Reviewer rymU (1/2)**
>
> **We sincerely thank you for your valuable suggestions. Here are our responses to your questions:**
>
> >**Q1. Performing spatio-temporal fusion in the 3D scaffold domain:**
>
> Omni-Scene combines a 2D image-level fusion with a 3D triplane fusion for spatial consistency, but does not incorporate temporal fusion across frames. In contrast, UniSplat performs both spatial and temporal fusion directly within a unified 3D latent scaffold. This design offers two key advantages.
>
> **Efficiency.** Image-domain fusion (e.g., via cross-attention) typically scales quadratically with the number of input views and frames. By lifting features into a unified 3D scaffold, our approach decouples fusion cost from input view count. The runtime of our fusion module grows slowly as we increase the number of views from 3 to 9:
> | #Views  | 3  | 5  | 7  | 9 |
> |-------:|---:|---:|---:|----:|
> | Fusion time (ms) | 28 | 55 | 62 | 71 |
>
> **Accuracy in dynamic modeling.** Fusing features in an ego-centric 3D scaffold naturally separates ego-motion from object motion, which is difficult to disentangle directly in the 2D image domain. Consequently, UniSplat ensures more stable temporal aggregation, leading to improved dynamics learning and consistent reconstructions over time.
>
> **Unified fusion.** Unifying spatial and temporal fusion within a single 3D scaffold keeps the overall architecture simple and conceptually coherent, allowing information from current views and historical frames to interact within the same space. This fused scaffold also serves as a streaming memory, supporting the aggregation of the longer temporal context.
>
> >**Q2. Design of the dual-branch decoder:**
>
> We design the dual-branch decoder to exploit the complementary strengths of point-anchored and voxel-generated Gaussians:
>
> **Point decoder.**  Guided by the geometry foundation model, point-anchored Gaussians are effective at recovering high-frequency details and sharp textures. However, relying solely on sparse points often leads to rendering artifacts in novel views.
>
> **Voxel decoder.** This branch generates Gaussians from voxel centers, which act as a skeleton of the scene and help fill in regions with insufficient point coverage, thereby enhancing scene completeness. As shown in **Table 5 of the main paper,** removing the voxel branch noticeably degrades LPIPS (from 0.30 to 0.38). **Figure 7 (Appendix Sec A.3, page 17)** further visualizes the effects of using only point Gaussians, only voxel Gaussians, and both together.
>
> **Fusion strategies.** The two branches already interact implicitly within the shared spatio-temporal scaffold, and at rendering time, we simply take the union of the two Gaussian sets. Explicit schemes, such as per-Gaussian adaptive weighting, are non-trivial in our setting because the two branches produce Gaussians with no one-to-one correspondence. Establishing stable pairwise weights would require computationally expensive matching or clustering steps. We believe exploring such sophisticated fusion at the primitive level is a new direction for future work.
>
>
> >**Q3. Robustness of the dynamic filtering threshold:**
>
> We observe that our method is largely insensitive to the specific choice of the dynamic filtering threshold. Concretely, we analyze two representative scenes with distinct motion patterns: walking persons (Scene A) and moving cars (Scene B). For each scene, we compare the dynamic-score distributions of moving objects versus static background:
>
> | Scene            | Region          | Mean Dynamic Score |
> |------------------|-----------------|--------------------|
> | Scene A   | Dynamic (person) | 0.89               |
> | Scene A   | Static background | 0.005               |
> | Scene B       | Dynamic (car)    | 0.95               |
> | Scene B       | Static background | 0.002               |
>
> In both scenes, the scores for dynamic regions are consistently high and clearly separated from the static background. This indicates that our learned score primarily captures the existence of motion rather than its magnitude, ensuring robustness against motion variability and sensor noise. Therefore, a simple fixed threshold is sufficient in our practice.
>
> * Scene A: segment-1105338229944737854_1280_000_1300_000 (frames 140–198);
>
> * Scene B: segment-17612470202990834368_2800_000_2820_000 (frames 0–100).

---

> > ### Author Response · Authors · 2025-11-21
> > **Response to Reviewer rymU (2/2)**
> >
> > >**Q4. Comparison with diffusion-based and token-based representations:**
> >
> > We thank the reviewer for this insightful question. Both lines of work are closely related to ours, and we have added a discussion to the revised Related Work section.
> >
> > **Diffusion-based.** Diffusion models can generate highly plausible details but inherently risk hallucinating content that is not strictly supported by the input views. In safety-critical autonomous driving, reconstruction must remain geometrically faithful to the sensor observations rather than merely visually realistic.
> >
> > **Token-based.** In token-based frameworks (e.g., LVSM [1]), each novel target view typically requires running a large transformer decoder conditioned on the input tokens. In contrast, UniSplat constructs a persistent explicit 3D representation. Once the scaffold and Gaussians are built, we can render arbitrary viewpoints via Gaussian rasterization alone (>100 FPS). Furthermore, our explicit 3D representations naturally support downstream tasks such as scene editing, which is considerably more difficult with implicitly entangled latent tokens.
> >
> >
> > >**Q5. Performance under extremely sparse coverage:**
> >
> > In typical autonomous driving configurations, cameras are mounted in spatially adjacent positions with limited mutual overlap. Consequently, reducing the number of views while retaining these adjacent cameras does not substantially compromise our pipeline, as the 3D scaffold can still maintain high-quality fusion within the regions covered by the remaining cameras.
> >
> > To verify this, we re-run UniSplat using only three front-facing cameras with the same training configuration as in our ablations. Under this setting, UniSplat achieves a novel-view PSNR of 26.68 dB, where "novel views" follow our main evaluation protocol and refer to frames at the next timestamp ($t+1$). The relatively high performance in this setting is partly due to the fact that these novel front-facing views are dominated by forward ego-motion, which poses fewer geometric challenges than the lateral shifts typical of side views.
> >
> >
> >
> > >**Q6. Robustness of streaming memory to misclassified dynamic objects:**
> >
> > Yes, there are scenarios where misclassified dynamic regions can lead to residual artifacts in the streaming memory. We have explicitly visualized and analyzed such a failure case in the last row of **Figure 5 (Appendix Sec A.3, page 15)**. In our current design, historical memory within the current Field of View (FOV) is explicitly discarded (refer to Eq. 9 in Sec. 3.4), thereby eliminating false positives in visible regions by prioritizing fresh reconstruction. However, artifacts located outside the current FOV may persist. Developing a "forgetting" or "correction" mechanism, which can clean up erroneous memory based on global consistency or semantic logic, is an important direction for future research.
> >
> >
> > [1] LVSM: A Large View Synthesis Model with Minimal 3D Inductive Bias, ICLR 2025

---

> > > ### Comment · Reviewer_rymU · 2025-11-26
> > >
> > > The author's rebuttal resolved most of my concerns, and I will maintain my rating.

---

### Official Review · Reviewer_cKtV · 2025-11-01

**Soundness:** 3
**Presentation:** 3
**Contribution:** 2
**Rating:** 4
**Confidence:** 5

**Summary:**

The paper proposes UniSplat, a feed-forward framework for dynamic driving-scene reconstruction that (i) builds an ego-centric 3D latent scaffold by fusing features from a geometry foundation model and a visual foundation model, (ii) performs spatio-temporal fusion directly in the scaffold via sparse 3D UNets with pose-warped accumulation, and (iii) decodes dynamic-aware Gaussians using a dual branch (point-anchored + voxel-anchored) with a streaming static-memory to complete unseen regions.

The dynamic prediction is supervised by pseudo-labels. These labels for moving objects are identified via 3D bounding-box tracking; the boxes are projected to the image to prompt SAM2, which returns dynamic segmentation masks.

On Waymo and nuScenes, UniSplat reports leading image metrics for reconstruction and novel view synthesis, and claims robustness for viewpoints outside camera coverage.

**Strengths:**

- An engineering-first pipeline. This paper proposes three-stage pipeline with a good motivation for 3D scaffold–space fusion, using sparse 3D UNets for spatial aggregation and pose-conditioned temporal accumulation. Overall it's easy to follow.
- Dynamic-handling. The proposed method can handle dynamic scenes.
- Experimental coverage. Evaluations on Waymo and nuScenes with both quantitative tables and qualitative figures; ablations cover feature composition, spatial vs temporal fusion, and decoder branches.

**Weaknesses:**

- **A System with Limited Conceptual Novelty**: The primary weakness is the paper's contribution, which appears to be more of a strong engineering effort than a conceptual breakthrough.
  - The “3D latent scaffold” (spatial-fusion) is essentially a sparse voxel grid with fused geometry+semantic features, a very standard way (fused in 3D grids) for fusing 3D features in many domains (like SLAM, or 3D understanding) and recent generalizable 3DGS/voxel/triplane works.
  - Warping and fusing features/primitives from previous frames by a pose-warped addition is another straightforward and known method used by the community for years (e.g., BEVFormer, BEVFusion and earlier works).
  - Using powerful, frozen geometry and visual foundation models to provide strong priors is a common practice.
  - The dual point/voxel decoder branches are a sensible design but I feel this is a compromise to sparse-voxel representations.
  - While the final system is effective, the core ideas themselves are largely incremental, making the work feel less insightful and more engineering-centric.

- Missing Related Work & Supervision Dependency: The paper misses discussion of several highly relevant works on feed-forward dynamic scene reconstruction, such as Flux4D (Want et al., 2025) and STORM (Yang et al., 2025). These methods also tackle large-scale, dynamic outdoor scenes, but they explore unsupervised 4D reconstruction, i.e., they do not require dynamic segmentation masks from external sources like object bounding boxes and SAM2.


- (Not a weakness) Setting novelty aside, this is a well-executed project with sufficient engineering efforts and clear implementation details.


[1] Want et al. "Flux4D: Flow-based Unsupervised 4D Reconstruction." In NeurIPS 2025

[2] Yang et al. "STORM: Spatio-Temporal Reconstruction Model for Large-scale Outdoor Scenes." In ICLR 2025

**Questions:**

1. Positioning / novelty. Is there any unique capability or perspective/insight offered by the proposed method?

2. Baseline Strength vs. Main Results. I understand that numbers in ablation studies are not directly comparable to those in main results, but it seems like a minimal UniSplat variant (without several add-ons) already performs strongly, possibly already exceeding the competitor results in the main table?

3. End-to-end latency. Please report full inference latency that includes the geometry/vision foundation stages.

4. External-Free and Voxel-Only Variants. Are these experiments feasible in the current setting?
  - Table 3: Add a no-Geo / no-Sem point (no external models) to show the fully end-to-end trained performance.
  - Table 5: Add a voxel-only prediction variant (no point-anchored branch) to isolate the value of the dual-branch decoder.

---

> ### Author Response · Authors · 2025-11-21
> **Response to Reviewer cKtV (1/2)**
>
> **We sincerely thank you for your valuable suggestions. Here are our responses to your questions:**
>
> >**W1. Clarification on novelty and technical contributions:**
>
> We thank the reviewer for the thoughtful comments. While UniSplat builds upon several established techniques in 3D vision, its core contribution is the design of a coherent, feed-forward framework in which geometric priors, semantic cues, temporal fusion, and persistent memory operate jointly through a unified latent 3D scaffold, rather than the introduction of completely new primitives.  This design enables real-time, temporally coherent reconstruction of dynamic driving scenes in a single pass.
>
> **On the comparison to standard voxel grids.** UniSplat instantiates the scaffold as a sparse voxel grid, but it is not used merely for spatial fusion. In our framework, the scaffold functions as a persistent 3D latent memory that is continuously updated with multi-frame geometric and semantic cues, supports dynamic–static separation and actor extraction, and decodes Gaussians for rendering. In principle, the same formulation could also be instantiated with other structured 3D representations, such as BEV or triplanes.
>
> **On pose-warped temporal fusion.** While warping is a known operation, UniSplat innovates by applying it within a streaming 3D latent scaffold tailored to driving scene reconstruction. Fusion within this metric scaffold effectively consolidates spatio-temporal context. This allows the model to implicitly disentangle dynamic objects from static content, ensuring consistent temporal updates and resulting in improved completeness and robust reconstruction.
>
> **On the use of foundation models.** Using strong pretrained priors is indeed increasingly common, but our contribution lies in how these heterogeneous priors are fused into a single evolving 3D latent scaffold. The geometry prior guides structural consistency, while the semantic prior enriches object awareness and scene understanding. Their interaction inside the scaffold is key to achieving high-quality dynamic reconstruction in a feed-forward manner.
>
> **On the dual decoder branches.** The dual-branch design reflects two complementary aspects of reconstruction: robust structural geometry and fine view-dependent detail. Integrating both pathways allows the system to achieve more complete and robust reconstructions in dynamic outdoor scenes. A visual comparison of the two branches is provided in **Figure 7 (Appendix Sec A.3, page 17).**
>
> **Summary.** UniSplat is not merely a collection of components, but a cohesive architectural paradigm that unifies geometry, semantics, temporal fusion, and persistent memory within a shared 3D latent scaffold.  This integration enables a feed-forward reconstruction of dynamic driving scenes with temporally coherent dynamics and stable static context, which has been difficult for prior pipelines to achieve. We hope this clarification highlights the contribution and significance of our work.
>
> >**W2. Missing related work & Supervision dependency:**
>
> We thank the reviewer for pointing out these missing discussions, and we have revised the paper to explicitly include them in the Related Work section.
>
> STORM focuses on unsupervised 4D reconstruction and scene flow prediction from multi-frame image inputs using a 2D Transformer. In contrast, UniSplat performs spatio-temporal fusion within a 3D latent scaffold, which effectively handles spatial fusion under minimally overlapping views and models complex temporal scene motion. Specifically, operating in a unified 3D space naturally aligns spatially corresponding information from different views and enables an explicit separation of ego-motion and object motion. Recent work, Flux4D, also employs a 3D network for dynamic modeling, but it requires LiDAR points for initialization.
>
> **Supervision strategy.** We agree that our method utilizes external models for dynamics learning, but we would like to clarify both the cost and motivation of our design:
>
> * **Cost.** Our dynamic segmentation masks are generated from 3D boxes (either provided by the datasets or obtained via offline detectors) and SAM2 as an off-the-shelf segmenter, making the pipeline practical for large-scale datasets without additional manual annotation.
>
> * **Motivation.** We believe that direct mask supervision can yield more stable and explicit static/dynamic separation, which is crucial for our static Gaussian memory. We also note that self-supervised motion learning strategies, such as those employed in STORM, are complementary to our framework and could be integrated as a complementary training signal in future work.

---

> > ### Author Response · Authors · 2025-11-21
> > **Response to Reviewer cKtV (2/2)**
> >
> > >**Q1. Positioning / novelty:**
> >
> > Please refer to W1.
> >
> > >**Q2. Baseline strength vs. Main results:**
> >
> > We thank the reviewer for this observation and agree that our baseline is relatively strong. It benefits from powerful foundation models pretrained on large-scale data, which provide strong priors for both geometry and appearance.
> >
> > **Clarification: Ablation vs. Main table settings.** A key difference is that the ablations in Table 4 use ground-truth scale initialization, while the main results do not. To provide a fair comparison under the main-table protocol, we instantiate a **Pixel-only baseline** corresponding to the first row of Table 4 but **without** GT scale initialization, and train it on the full training set. We compare it against DepthSplat and our UniSplat:
> >
> > | Model               | PSNR ↑ | SSIM ↑ | LPIPS ↓ | FID ↓ (lane-shift ±0.5 m) |
> > |---------------------|:------:|:------:|:-------:|:--------------------------:|
> > | DepthSplat | 23.86   | 0.70   | 0.31    | -
> > | Pixel-only baseline | 24.64  | 0.72   | 0.29    | 28.88                      |
> > | UniSplat       | **25.12**  | **0.74**   | **0.27**    | **24.28**                      |
> >
> > UniSplat brings consistent improvements over the strong Pixel-only baseline. To further assess the robustness under more severe viewpoint deviations, we conduct an evaluation in a lateral lane-shift configuration (±0.5 m) on 5 randomly selected Waymo validation scenes (front camera only). The benefit of UniSplat becomes more pronounced, with **-4.60 FID reduction** over the Pixel-only baseline. This supports our claim that the proposed 3D scaffold and streaming memory are particularly important for robust reconstruction and completion.
> >
> > >**Q3. End-to-end latency:**
> >
> > On a single NVIDIA H20 GPU, our pipeline processes 5 input images at 350×518 resolution in ~424 ms: ~180 ms for the geometry/vision foundation stage, ~220 ms for the UniSplat core (3D scaffold construction, spatio-temporal fusion, and Gaussian generation), and ~24 ms to render 10 target views.
> >
> > >**Q4. External-free and voxel-only variants:**
> > We thank the reviewer for the suggestion and have evaluated the two variants.
> >
> > **Table 3: External-free variant.** We trained a variant initialized from scratch, removing all external geometry and semantic foundation models. To facilitate geometric learning, we added an auxiliary depth prediction loss. This model converges but shows clearly degraded performance compared to the foundation-based UniSplat:
> > | Geo | Sem | PSNR ↑ | SSIM ↑ | LPIPS ↓ |
> > |:---:|:---:|:------:|:------:|:-------:|
> > | ✗   | ✗   | 24.46  | 0.70   | 0.35    |
> > | ✓   | ✓   | **25.08**  | **0.74**   | **0.30**    |
> >
> > **Table 5: Voxel-only variant.** We also implemented a voxel-only decoder that removes the point-anchored branch and predicts scene content purely within the voxel grid. Given its lower spatial granularity relative to point-based features, the voxel-only variant naturally exhibits limited performance due to blurrier reconstructions and the loss of distant structures. Even so, it contributes a coarse-grained geometric backbone that complements the finer details captured by the point branch, as demonstrated in **Figure 7 (Appendix Sec A.3, page 17).**
> > | Point | Voxel | PSNR ↑ | SSIM ↑ | LPIPS ↓ |
> > |:------------:|:------------:|:------:|:------:|:-------:|
> > | ✗            | ✓            | 21.51  | 0.60   | 0.54    |
> > | ✓            | ✗            | 24.62  | 0.72   | 0.38    |
> > | ✓            | ✓            | **25.08**   | **0.74**   | **0.30**    |

---

> > > ### Comment · Reviewer_cKtV · 2025-11-24
> > >
> > > I thank authors for detailed rebuttals. From the rebuttals, I see the benefits from foundation models and the necessity of the voxel decoder branch. Although my concern about novelty is not fully addressed by the rebuttals, I appreciate the efforts made by authors and believe that this could be a good system.
> > >
> > > Based on these, I will increase my rating to 6.

---

### Official Review · Reviewer_uYox · 2025-11-06

**Soundness:** 3
**Presentation:** 3
**Contribution:** 3
**Rating:** 6
**Confidence:** 4

**Summary:**

This paper presents UniSplat, a feedforward framework for dynamic driving scene reconstruction and novel view synthesis. The core idea is to construct a 3D latent scaffold that unifies multi-view spatial and multi-frame temporal fusion. The scaffold is built by leveraging geometry and visual foundation models to encode geometry and semantics in 3D space. UniSplat then introduces a Gaussian decoder to generate the 3DGS reconstruction from the scaffold, and a memory mechanism to maintain the accumulated static Gaussians over time to improve completion. The proposed UniSplat achieves state-of-the-art performance on Waymo and nuScenes datasets and demonstrates SoTA performance for novel view synthesis compared to existing feed-forward reconstruction methods.

**Strengths:**

* The paper is well written and clearly structured, making the technical ideas easy to follow despite the method’s complexity.
* The proposed scaffold-based fusion mechanism is both intuitive and practical, effectively addressing the challenge of modelling dynamic scenes while supporting progressive memory updates.
* Experimental evaluation is thorough, including comparisons to multiple strong baselines on two large-scale driving datasets, with consistent quantitative and qualitative improvements. Ablation studies also provide convincing evidence of the contribution of each component.

**Weaknesses:**

* It is not entirely clear how the scaffold-based fusion mechanism handles dynamic content. For example, when a vehicle moves across frames, how are its features aligned or updated consistently during fusion?
* The paper shows novel view rendering by rotating the camera, but it remains unclear whether dynamic actors can be moved or manipulated. Supporting such editing would enable full camera simulation.
* Maintaining and updating a dense 3D scaffold could be computationally expensive for long sequences or large-scale scenes. The paper does not analyze memory usage or runtime trade-offs beyond short sequences.
* The evaluation focuses mainly on reconstruction metrics. Additional experiments on temporal consistency, and robustness under occlusions would make the claims more convincing.
* The predicted dynamic masks appear misaligned with RGB images in the supplementary video (e.g., at 1:03). Clarifying the source of this misalignment would strengthen the work.

**Questions:**

* How does UniSplat perform when extended beyond driving scenes to other dynamic 4D environments (e.g., human motion, indoor scenes)?
* Could the authors clarify the scalability of the scaffold fusion to longer sequences (e.g., 1–2 minutes) and higher-resolution scenes?
* Can the streaming memory lead to error accumulation or stale Gaussians over time? How is this mitigated?

---

> ### Author Response · Authors · 2025-11-21
> **Response to Reviewer uYox (1/3)**
>
> **We sincerely thank you for your valuable suggestions. Here are our responses to your questions:**
>
> >**W1. How does the scaffold-based fusion mechanism handle dynamic content?**
>
> We thank the reviewer for this insightful question. Our scaffold-based fusion handles dynamic content through implicit learning: by jointly observing multi-frame features, the model learns how dynamic signals should be fused in the latent space.
>
> Although features from moving objects are aggregated into scaffold voxels like other content, the training objective guides the model to implicitly reduce the influence of outdated voxel evidence. As a result, historical locations of a moving object naturally contribute very little during rendering, while the model focuses on the object’s most recent, geometrically consistent observations. This effect is clearly shown in **Figure 6 (Appendix Sec A.3, page 17):** as another vehicle moves across the scene, the voxel-rendered results contain no ghosting artifacts or temporal inconsistencies, demonstrating that the scaffold learns to downweight old dynamic content and maintain a clean, temporally consistent reconstruction.
>
> >**W2. Can dynamic actors be moved or manipulated to enable full camera simulation?**
>
> Yes, our framework can support manipulation of dynamic actors, enabled by the learned dynamic representations and the memory mechanism in our model.
>
> **Extracting dynamic actors.** Using the learned dynamic scores, we first render a 2D dynamic mask, extract connected components, and then gather all Gaussians that both fall inside a selected region and exhibit sufficiently high dynamic scores. This reliably isolates a specific moving actor that can be removed, moved, or reinserted.
>
> **Restoring the background after manipulation.** When a dynamic actor is removed or relocated, regions of the static background that it currently occludes become visible and would appear as empty holes if only the current frame were used. Our memory mechanism (Sec. 3.4) addresses this by maintaining static Gaussians accumulated from previous viewpoints, including those not visible in the current frame. Specifically, in Eq. 9 and Eq. 10 of the main paper, we utilize the full memory state $\mathcal{M}\_{t-1}$ rather than the filtered state $\mathcal{M}\_{t-1}^{\prime}$ to reconstruct these previously occluded background regions, while filtering redundant static Gaussians for consistency.
>
> **Unified re-rendering with edited actors.** After extraction and background restoration, the edited actor (e.g., moved to a new position) can be directly combined with the reconstructed static scene, and both are rendered together as a unified Gaussian field. This enables coherent removal, relocation, and insertion of dynamic actors.
>
> As shown in **Figure. 8 (Appendix Sec A.4, page 18),** we demonstrate examples of removing, moving and adding dynamic actors.
>
> >**W3. About memory and runtime efficiency:**
>
> On Waymo sequences (~20 seconds, 200 frames), our 3D scaffold construction and spatial-temporal fusion process takes 80-100ms per iteration, with memory usage maintained at 16-17GB throughout the sequence. This demonstrates that our method scales efficiently to long sequences without unbounded growth, enabled by two key strategies:
>
> **Spatial range constraint.** We limit the scaffold to a fixed spatial range around the current viewpoint. Historical voxels outside this range are automatically discarded during fusion.
>
> **Adaptive pruning.** Before fusing the historical scaffold with the current frame, we apply opacity-based pruning to retain only high-opacity voxels. Specifically, we compute each voxel's opacity by averaging the opacities of Gaussians generated and select the top 60% of voxels using a quantile threshold, preserving the most reliable content while discarding low-confidence regions.

---

> > ### Author Response · Authors · 2025-11-21
> > **Response to Reviewer uYox (2/3)**
> >
> > >**W4. Temporal consistency and robustness under occlusions:**
> >
> > **Robustness under occlusions.** As discussed in W2 and illustrated in Figure 8 (Appendix Sec A.4, page 18), the temporal memory mechanism enables UniSplat to faithfully recover background regions that are occluded by dynamic actors in the current frame. This demonstrates that the model maintains stable static content even when significant parts of the scene are temporarily hidden. We additionally provide more actor-removal examples in **Figure 9 (Appendix Sec A.4, page 18),** showing that the reconstructed backgrounds remain photorealistic and consistent with their surroundings, further validating the effectiveness of our approach under occlusion.
> >
> > **Temporal consistency.** Our 3D scaffold propagates historical context into current-frame reconstruction, providing strong temporal stability across time. We note, however, that for very long sequences, small misalignments may gradually accumulate in the streaming Gaussian memory. This is primarily because the current pipeline does not impose explicit long-horizon constraints to maintain global scale alignment. Incorporating multi-frame supervision or lightweight bundle-adjustment–style regularization is a promising direction for enhancing long-term consistency, and we plan to explore this in future work.
> >
> > >**W5. Dynamic mask misalignment in supplementary video:**
> >
> > Thank you for pointing this out. The perceived misalignment is actually due to **transparency effects from glass surfaces**. During the dynamic mask rendering process, the glass regions have low opacity in Gaussian representation, which allows the background dynamic object to be visible through alpha-blending, createing a visual appearance of misalignment between the dynamic mask and RGB content.
> >
> > We will add a clarification note in the supplementary material to explain this rendering behavior.
> >
> > >**Q1. Generalization to indoor scenes**
> >
> > We conducted preliminary experiments on the TUM-Dynamics dataset [1]. Following MonST3R [2], we use the 8 dynamic sequences with 3× temporal subsampling. We train on 7 sequences and test on the remaining 1. The model achieves **21.02 PSNR** on novel view synthesis.
> >
> > This performance is noticeably weaker than on large-scale driving datasets, which we attribute mainly to: (1) the limited training data (7 indoor sequences vs. thousands of driving scenes), and (2) the single-view input setting in TUM-Dynamics, whereas UniSplat is designed for multi-view surround setups that provide stronger geometric constraints. Extending UniSplat to single-view indoor dynamic scenes requires more dedicated investigation, which we leave for future work.
> >
> > >**Q2. Scalability to longer sequences and higher-resolution scenes:**
> >
> > We would like to clarify the scalability of our scaffold fusion mechanism in both temporal and spatial dimensions.
> >
> > **Longer sequences.** As discussed in W3, our method maintains stable computational and memory costs regardless of sequence length through spatial range constraints and adaptive pruning. Therefore, extending from 20-second sequences to 1-2 minute sequences does not introduce additional computational burden.
> >
> > **Higher-resolution scenes.** We additionally evaluated our method at 2× input image resolution. Since the fusion operates entirely in 3D voxel space rather than image space, the computational complexity grows only mildly with increased pixel resolution. In practice, the scaffold-fusion cost rises only from ~55 ms to ~76 ms per frame, confirming that the method scales well to higher-resolution inputs.
> >
> > [1] A benchmark for the evaluation of RGB-D SLAM systems, IROS 2012
> >
> > [2] MonST3R: A Simple Approach for Estimating Geometry in the Presence of Motion, ICLR 2025

---

> > > ### Author Response · Authors · 2025-11-21
> > > **Response to Reviewer uYox (3/3)**
> > >
> > > >**Q3. Error accumulation and stale Gaussians in streaming memory:**
> > >
> > > We thank the reviewer for raising this important question. Error accumulation is indeed a potential concern in streaming reconstruction, and our system mitigates it through different mechanisms in the two memory components.
> > >
> > > * **Latent scaffold memory:** The latent scaffold is continuously updated on every frame using fresh feature projections. This provides a natural self-correction mechanism: newly observed evidence overwrites or refines previously accumulated features, preventing long-term drift. In practice, this per-frame refinement keeps the scaffold representation well aligned with the current scene geometry.
> > >
> > > * **Explicit Gaussian memory $\mathcal{M}_{t}$:** We currently do not apply a dedicated error-correction module to the explicit Gaussian memory. However, its susceptibility to error accumulation is limited in practice because $\mathcal{M}_{t}$ mainly stores Gaussians corresponding to regions that fall outside the current field of view. These Gaussians are not redundant or outdated; they capture valid parts of the scene that are simply not visible in the current frame. When those regions re-enter the field of view (FOV), the scaffold-based fusion updates them again using current observations, preventing the memory from becoming stale.
> > >
> > > **Stale Gaussians.** We emphasize that the "stale" Gaussians in our streaming memory are not redundant but rather retain meaningful scene content. Since our memory primarily stores Gaussians from regions outside the current FOV, these historical Gaussians represent valid observations of previously visible areas rather than outdated duplicates.

---

### Author Response · Authors · 2025-12-03
**Summary of the Rebuttal**

We appreciate the reviewers' thoughtful feedback and constructive suggestions, which have helped us improve the clarity and completeness of our manuscript. **All reviews indicate a positive assessment of our system and its practical relevance for dynamic driving scene reconstruction.** The key points of their feedback are summarized below:
* **Reviewer uYox** affirmed that our scaffold-based fusion, combined with the streaming memory, effectively addresses the complexities of dynamic driving scenarios, while praising our strong performance on two large-scale driving datasets.
* **Reviewer cKtV**, while initially raising concerns about conceptual novelty, recognized the effectiveness and soundness of our system design. Following our clarifications, the reviewer acknowledged the benefits of our foundation model integration and dual-branch decoder, **thereby raising the rating to 6 on November 24.**
* **Reviewer rymU** acknowledged our unified spatio-temporal fusion approach within a single 3D scaffold, highlighting that our memory streaming mechanism effectively addresses ghosting from dynamic objects while enabling comprehensive scene completion.
* **Reviewer 2VZv** recognized that our 3D latent scaffold provides a principled framework for fusing multi-view and temporal information beyond prior 2D aggregation methods, and noted that our dynamic-aware Gaussian memory enables persistent scene reconstruction while reducing ghosting artifacts.

We have addressed each reviewer's concerns individually in our detailed responses. We also revised our manuscript with the following major additions:
1. We expanded the Related Work section with **discussion on unsupervised 4D reconstruction methods and comparisons with diffusion-based and token-based representations** (cKtV-W2, rymU-Q4).
2. We added visualization of **voxel-rendered results in dynamic scenes in Figure 6,** confirming our scaffold-based fusion capabilities in handling dynamic content (uYox-W1).
3. We included examples of **actor-level editing capabilities in Figure 8 and Figure 9,** along with a detailed pipeline description in Appendix Sec. A.4 (uYox-W2, uYox-W4).
4. We provided **separate visualizations of each decoder branch in Figure 7,** demonstrating the complementary roles of point-based and voxel-based reconstruction (cKtV-Q4, rymU-Q2, 2VZv-W3).

---

### Meta-Review · Area_Chair_u8gW · 2026-01-07

**Summary:**

All four reviewers find the contribution of this work meaningful for dynamic driving-scene reconstruction and consider this system technically solid.

Prior to rebuttal, most scores center around the borderline accept. The main risk for rejection was the perception of limited conceptual novelty (raised by Reviewers cKtV and rymU), along with requests to clarify key methodological details and strengthen ablation. After rebuttal, reviewers converged to a positive borderline acceptance consensus.

The AC has carefully reviewed the paper, prior-rebuttal reviews, rebuttals, and discussion, and agrees with this assessment. The final recommendation is Accept (Poster).

**Reviewer Concerns:**

The major concerns addressed by the rebuttal:
- Additional ablations (by reviewers cKtV, 2VZv, uYox). The rebuttal provides the required ablations that include but not limited to (a) a dependency-free variant that avoids the use of foundation models, (b) a voxel-only decoder variant that shows why the dual-branch strategy is helpful, and (c) run-time and memory analysis. These ablations strengthen the comprehensiveness of this work.

- Methodological clarity (by reviewers uYox, rymU, 2VZv). The rebuttal provides clearer explanations and additional visualizations regarding dynamic handling, the role of the dual-branch decoder, and the behavior of the memory mechanism.

- Novelty (by reviewers cKtV and rymU). The primary concern that was not fully addressed is novelty. The AC agrees the novelty concerns raised by multiple reviewers, but also agrees that the overall system design constitutes a meaningful contribution to the field.

Outstanding issues are mostly about testing the generalizability of the proposed method to longer sequences and in-door scenes, which the authors acknowledge as future work rather than fully resolving in this submission.

**Reviewer Scores:**

- Reviewer cKtV explicitly mentioned increasing score to 6.
- Reviewers rymU and 2VZv explicitly mentioned maintaining their scores.
- Reviewer uYox did not engage in the discussion, but I expect the reviewer would maintain the original positive assessment since most concerns were addressed in the rebuttal.

---

### Decision · Program_Chairs · 2026-01-26

Accept (Poster)